# Learning in complex action spaces without policy gradients

**Arash Tavakoli**                                                                 *atavakoli@riotgames.com*
*Riot Games*

**Sina Ghiassian**                                                                      *sinag@spotify.com*
*Spotify*

**Nemanja Rakićević**                                                                 *rakicevic@google.com*
*Google DeepMind*

**Reviewed on OpenReview:** *https: // openreview. net/ forum? id= nOL9M6D4oM*

## Abstract

While conventional wisdom holds that policy gradient methods are better suited to complex action spaces than action-value methods, foundational work has shown that the two paradigms are equivalent in small, finite action spaces (O'Donoghue et al., 2017; Schulman et al., 2017a). This raises the question of why their computational applicability and performance diverge as the complexity of the action space increases. We hypothesize that the apparent superiority of policy gradients in such settings stems not from intrinsic qualities of the paradigm but from universal principles that can also be applied to action-value methods, enabling similar functions. We identify three such principles and provide a framework for incorporating them into action-value methods. To support our hypothesis, we instantiate this framework in what we term QMLE, for Q-learning with maximum likelihood estimation. Our results show that QMLE can be applied to complex action spaces at a computational cost comparable to that of policy gradient methods, all without using policy gradients. Furthermore, QMLE exhibits strong performance on the DeepMind Control Suite, even when compared to state-of-the-art methods such as DMPO and D4PG. We make our code publicly available.

## 1 Introduction

In reinforcement learning, policy gradients (and their actor-critic variants) (Sutton et al., 1999) have become the backbone of solutions for environments with complex action spaces (Dulac-Arnold et al., 2015; OpenAI et al., 2019; Vinyals et al., 2019; Ouyang et al., 2022). We use the term "complex action spaces" broadly to refer to those for which enumerating actions or brute-force operations are computationally intractable, such as domains with multi-dimensional continuous or high-dimensional discrete actions (Hubert et al., 2021). In contrast, action-value methods—primarily, variants of Sarsa and Q-learning—have traditionally been confined to tabular-action models for small and finite action spaces. However, where applicable, such as on the Atari Suite (Bellemare et al., 2013; Machado et al., 2018), action-value methods are frequently the preferred approach over policy gradient methods (Kapturowski et al., 2023; Schwarzer et al., 2023).

In recent years, foundational research has shown that the distinction between action-value and policy gradient methods is narrower than previously understood, particularly in the basic case of tabular-action models in small and finite action spaces (see, e.g., Schulman et al., 2017a). Notably, O'Donoghue et al. (2017) established a direct equivalence between these paradigms, connecting the fixed-points of action-preferences of policies optimized by regularized policy gradients to action-values learned by action-value methods. These insights invite further exploration of discrepancies emerging as action-space complexity increases.

*What core principles underpin the superior computational applicability and performance of policy gradient methods in such settings?* In this paper, we identify three such principles. First, policy gradient methods leverage Monte Carlo (MC) approximations for summation or integration over action spaces, enabling

computational feasibility even in environments with complex action spaces. Second, they employ amortized maximization through a special form of maximum likelihood estimation (namely, the policy gradient itself), iteratively refining the policy to favor high-value actions without requiring brute-force arg max over the action space. Third, scalable policy gradient methods employ action-in architectures for action-value approximation, implicitly facilitating representation learning and generalization across the joint state-action space.

*Are these principles exclusive to policy gradient methods?* We argue that these principles can be adapted to action-value methods, thereby bridging the scalability and performance gap between the two paradigms. Instead of using MC methods for summation or integration as in policy gradient methods, we propose using MC methods to approximate the arg max operation, making action-value methods computationally scalable for complex action spaces. Moreover, explicit maximum likelihood estimation can enable caching and iterative refinement of parametric predictors for amortized arg max approximation. Lastly, action-in architectures can be employed not only to scalably evaluate a limited set of actions in any given state, but also to enable representation learning and generalization across both states and actions.

To test our arguments, we introduce *Q-learning with maximum likelihood estimation* (QMLE). Given that QMLE builds upon Q-learning, DDPG (Lillicrap et al., 2016) serves as its closest counterpart among commonly used policy gradient methods for continuous-action tasks, and thus is our primary baseline for comparison.

Through systematic ablation studies (§5.3), we show that each identified principle individually contributes to improved scalability or performance of action-value learning. Benchmarking QMLE against DDPG on diverse multi-dimensional continuous control tasks, ranging from 1 to 21 action dimensions, we demonstrate that incorporating all three principles collectively enables QMLE to achieve competitive performance (§5.2). These results provide evidence that the identified principles are indeed fundamental to the broad applicability of policy gradient methods in complex action spaces, and that they are not intrinsically tied to the policy gradient paradigm and can be effectively adapted into action-value learning.

Beyond multi-dimensional continuous-action scenarios, we demonstrate the applicability of QMLE to large, high-dimensional discrete-action ones by employing a 3-bin discretization scheme on our continuous control benchmarks. Our results illustrate QMLE's robustness in managing increasing complexity due to the curse of dimensionality, effectively scaling from 2 to 38 action dimensions (corresponding to discrete action sets ranging from only $3^2 = 9$ up to $3^{38}$, approximately 1.35 quintillion, actions).

The idea of using sampling-based approximation of the arg max in value-based methods has been explored in earlier works. For example, Tian et al. (2022) studied the combination of value iteration and random search in discrete domains, with a tabular mechanism for tracking the best historical value-maximizing action in each state. Kalashnikov et al. (2018) introduced the QT-Opt algorithm, which employs a fixed stochastic search via the cross-entropy method to approximate arg max in Q-learning. Closely related to QMLE is the AQL algorithm by de Wiele et al. (2020), which integrates Q-learning with entropy-regularized MLE to approximate a value-maximizing action distribution. While QMLE shows superior performance relative to QT-Opt and AQL in complex action spaces (§C), our emphasis in this work is less on algorithmic novelty and more on dissecting the core principles that bridge the gap between the two paradigms.

## 2 Background

### 2.1 The reinforcement learning problem

The reinforcement learning (RL) problem (Sutton & Barto, 2018) is generally described as a Markov decision process (MDP) (Puterman, 1994), defined by the tuple $\langle \mathcal{S}, \mathcal{A}, \mathcal{P}, \mathcal{R} \rangle$, where $\mathcal{S}$ is a state space, $\mathcal{A}$ is an action space, $\mathcal{P} : \mathcal{S} \times \mathcal{A} \rightarrow \Delta(\mathcal{S})$[1] is a state-transition function, and $\mathcal{R} : \mathcal{S} \times \mathcal{A} \times \mathcal{S} \rightarrow \Delta(\mathbb{R})$ is a reward function. The behavior of an agent in an RL problem can be formalized by a policy $\pi : \mathcal{S} \rightarrow \Delta(\mathcal{A})$, which maps a state to a distribution over actions. The value of state $s$ under policy $\pi$ may be defined as the expected

---

[1]$\Delta$ denotes a distribution.

discounted sum of rewards: $V^\pi(s) \doteq \mathbb{E}_{\pi,\mathcal{P},\mathcal{R}}[\sum_{t=0}^\infty \gamma^t r_{t+1}|s_0 = s]$, where $\gamma \in (0,1)$ is a discount factor used to exponentially decay the present value of future rewards.[2]

The goal of an RL agent is defined as finding an optimal policy $\pi^*$ that maximizes this quantity across the state space: $V^{\pi^*} \geq V^\pi$ for all $\pi$. While there may be more than one optimal policy, they all share the same state-value function: $V^* = V^{\pi^*}$. Similarly, we can define the value of state $s$ and action $a$ under policy $\pi$: $Q^\pi(s,a) \doteq \mathbb{E}_{\pi,\mathcal{P},\mathcal{R}}[\sum_{t=0}^\infty \gamma^t r_{t+1}|s_0 = s, a_0 = a]$. Notice that the goal can be equivalently phrased as finding an optimal policy $\pi^*$ that maximizes this alternative quantity across the joint state-action space: $Q^{\pi^*} \geq Q^\pi$ for all $\pi$. Same as before, optimal policies share the same action-value function: $Q^* = Q^{\pi^*}$.

The state and action value functions are related to each other via: $V^\pi(s) = \sum_a Q^\pi(s,a)\pi(a|s)$, where we use $\sum$ to signify both summation and integration over discrete or continuous actions. For all MDPs there is always at least one deterministic optimal policy, which can be deduced by maximizing the optimal action-value function: $\arg\max_a Q^*(s,a)$ in any given state $s$. It is worth noting that there may be cases where multiple actions yield the same maximum value, resulting in ties. By breaking such ties at random, considering all conceivable distributions, we can construct the set of all optimal policies, including both deterministic and stochastic policies. Regardless of the optimal policy, the optimal state-value and action-value functions are related to each other in the following way: $V^*(s) = \max_a Q^*(s,a)$. Similarly, the optimal state-value function can be used to extract optimal policies by invoking the Bellman recurrence: $\arg\max_a \mathbb{E}_{\mathcal{P},\mathcal{R}}[r_{t+1} + \gamma V^*(s_{t+1})|s_t = s]$. However, this requires access to the MDP model, rendering the sole optimization of state-values unsuitable for model-free RL.

## 2.2 Action-value learning

Optimizing the action-value function and deducing an optimal policy from it seems to be the most direct approach to solving the RL problem in a model-free manner. To this end, we first consider the Bellman recurrence for action-values (Bellman, 1957):

$$Q^\pi(s,a) = \mathbb{E}_{\pi,\mathcal{P},\mathcal{R}}[r_{t+1} + \gamma Q^\pi(s_{t+1},a_{t+1})|s_t = s, a_t = a], \tag{1}$$

where $\pi$ is in general a stochastic policy and $a_{t+1} \sim \pi(.|s_{t+1})$. By substituting policy $\pi$ with an optimal policy $\pi^*$ and invoking $Q^*(s, \arg\max_a Q^*(s,a)) = \max_a Q^*(s,a)$, we can rewrite Eq. 1:

$$Q^*(s,a) = \mathbb{E}_{\mathcal{P},\mathcal{R}}[r_{t+1} + \gamma \max_{a'} Q^*(s_{t+1},a')|s_t = s, a_t = a]. \tag{2}$$

The method of temporal differences (TD) (Sutton, 1988) leverages equations (1) and (2) to contrive two foundational algorithms for model-free RL: Sarsa (Rummery & Niranjan, 1994) and Q-learning (Watkins, 1989). Sarsa updates its action-value estimates, $Q(s_t, a_t)$, by minimizing the TD residual:

$$\left(r_{t+1} + \gamma Q(s_{t+1}, a_{t+1})\right) - Q(s_t, a_t), \tag{3}$$

whereas Q-learning does so by minimizing the TD residual:

$$\left(r_{t+1} + \gamma \max_a Q(s_{t+1}, a)\right) - Q(s_t, a_t). \tag{4}$$

Both algorithms have been shown to converge to the unique fixed-point $Q^*$ of Eq. 2 under similar conditions, with one additional and crucial condition for Sarsa (Watkins & Dayan, 1992; Jaakkola et al., 1994; Singh et al., 2000). Namely, because Sarsa uses the action-value of the action chosen by its policy in the successor state, the action-values can converge to optimality in the limit only if it chooses actions greedily in the limit: $\lim_{k\to\infty} \pi_k(a|s) = \mathbf{1}_{a=\arg\max_{a'} Q(s,a')}$. This is in contrast with Q-learning which uses its maximum action-value in the successor state regardless of its policy, thus liberating its learning updates from how it chooses to act. This key distinction makes Sarsa an *on-policy* and Q-learning an *off-policy* algorithm. As a final point, the action-value function can be approximated by a parameterized function $Q$, such as a neural network, with parameters $\boldsymbol{\omega}$ and trained by minimizing the squared form of the TD residual (3) or (4).

---

[2]Discounts are occasionally employed to specify the true optimization objective, whereby they should be regarded as part of the MDP. However, more often discounts serve as a hyper-parameter (van Seijen et al., 2019).

### 2.3 Policy gradient methods

Unlike action-value methods (§2.2), policy gradient methods do not require an action-value function for action selection. Instead they work by explicitly representing the policy using a parameterized function $\pi$, such as a neural network, with parameters $\boldsymbol{\theta}$ and only utilizing action-value estimates to learn the policy parameters. To demonstrate the main idea underpinning policy gradient methods, we start from the following formulation of the RL problem (cf. §2.1):

$$\pi^* \doteq \arg\max_{\pi} \mathop{\mathbb{E}}_{\pi,\mathcal{P}} \left[ V^{\pi}(s_t) \right]. \tag{5}$$

The objective function in this formulation is the expected state-value function, where the expectation is taken over the state distribution induced by policy $\pi$ and state-transition function $\mathcal{P}$. This problem can be solved approximately via gradient-based optimization. In fact, this forms the basis of policy gradient methods. Accordingly, the policy gradient theorem (Sutton et al., 1999) proves that the gradient of the expected state-value function with respect to policy parameters $\boldsymbol{\theta}$ is governed by:

$$\nabla \mathop{\mathbb{E}}_{\pi,\mathcal{P}} \left[ V^{\pi}(s_t) \right] = \nabla \mathop{\mathbb{E}}_{\pi,\mathcal{P}} \left[ \sum_a Q^{\pi}(s_t, a)\pi(a|s_t) \right] \propto \mathop{\mathbb{E}}_{\pi,\mathcal{P}} \left[ \sum_a Q^{\pi}(s_t, a)\nabla\pi(a|s_t) \right]. \tag{6}$$

By using an estimator of the above expression, denoted $\widehat{\nabla J(\boldsymbol{\theta})}$, policy parameters can be updated via stochastic gradient ascent: $\boldsymbol{\theta} \leftarrow \boldsymbol{\theta} + \alpha\widehat{\nabla J(\boldsymbol{\theta})}$, where $\alpha$ is a positive step-size. It is important to note that, like Sarsa (§2.2), policy gradients are on-policy learners: applying one step of policy gradient updates the policy parameters $\boldsymbol{\theta} \to \boldsymbol{\theta}'$ and thereby the policy $\pi \to \pi'$, thus inducing a different action-value function $Q^{\pi} \to Q^{\pi'}$ and a different state distribution.

There have been attempts to extend policy gradients to off-policy data (Degris et al., 2012). The most common approach in this direction is to use deterministic policy gradients (DPG; Silver et al., 2014):

$$\nabla \mathop{\mathbb{E}}_{\pi,\mathcal{P}} \left[ V^{\pi}(s_t) \right] = \nabla \mathop{\mathbb{E}}_{\pi,\mathcal{P}} \left[ \int Q^{\pi}(s_t, a)\delta(a - \pi(s_t))\mathrm{d}a \right] \tag{7a}$$

$$= \nabla \mathop{\mathbb{E}}_{\pi,\mathcal{P}} \left[ Q^{\pi}(s_t, \pi(s_t)) \right] \tag{7b}$$

$$\propto \mathop{\mathbb{E}}_{\pi,\mathcal{P}} \left[ \nabla_a Q^{\pi}(s_t, a = \pi(s_t))\nabla\pi(s_t) \right]. \tag{7c}$$

This is similar to Eq. 6 with the difference that here we replace the general-form policy $\pi(a|s)$ with a deterministic and continuous policy $\delta(a - \pi(s))$, where $\delta$ denotes the delta function whose parameters are given by $\pi(s)$. Moreover, this derivation only holds in continuous action spaces and, as such, we substitute our general-form notation $\sum$ for both summation and integration with $\int$ to specify integration over continuous actions. The expression (7b) is then derived from (7a) by invoking the sifting property of the delta function and (7c) is deduced from (7b) by applying the chain rule, yielding a gradient with respect to actions (denoted $\nabla_a$) and another with respect to policy parameters $\boldsymbol{\theta}$ (denoted as before by the shorthand $\nabla$).

To implement an off-policy method using DPG, we must make two key changes to the true deterministic policy gradient (7). First, the deterministic policy—which is the target of optimization by DPG—generally differs from the behavior policy $\pi(a|s)$ that the agent uses to interact with and explore the environment. Therefore, we must modify our notation to reflect this distinction:

$$\mathop{\mathbb{E}}_{\pi,\mathcal{P}} \left[ \nabla_a Q^{\mu}(s_t, a = \mu(s_t))\nabla\mu(s_t) \right], \tag{8}$$

where $\mu$ denotes the parameters of the delta function $\delta$ and the expectation is computed with respect to the state distribution induced under behavior policy $\pi$ and state-transition function $\mathcal{P}$. Second, our estimator $Q \approx Q^{\mu}$ must be differentiable with respect to actions. This is typically achieved by training a parameterized function $Q$ by minimizing the squared form of the TD residual:

$$\left( r_{t+1} + \gamma Q(s_{t+1}, \mu(s_{t+1})) \right) - Q(s_t, a_t). \tag{9}$$

This expression can be viewed as substituting $Q(s_{t+1}, \mu(s_{t+1}))$ for $\max_a Q(s_{t+1}, a)$ in the TD expression (4), which is used by Q-learning.

## 2.4 Maximum likelihood estimation

Suppose we have a data set $\{(x_i, y_i)\}$ drawn from an unknown joint distribution $p(x, y)$, where random variables $x_i$ and $y_i$ respectively represent inputs and targets. Frequently, problem scenarios involve determining the parameters of an assumed probability distribution that best describe the data. The method of maximum likelihood estimation (MLE) addresses this by posing the question: "For which parameter values is the observed data most likely?". In this context, we typically start by representing our assumed distribution using a parameterized function $f$, such as a neural network, with parameters $\boldsymbol{\theta}$. Hence, $\boldsymbol{\phi} \doteq f(x)$ serves as our estimator for the distributional parameters in $x$. For example, $\boldsymbol{\phi}$ contains $K$ values in the case of a categorical distribution with $K$ categories, and contains means $\boldsymbol{\mu}$ and variances $\boldsymbol{\sigma}$ in the case of a multivariate heteroscedastic Gaussian distribution. We will denote the probability distribution that is specified by parameters $\boldsymbol{\phi} = f(x)$ as $f(y|x)$. The problem of finding the optimal parameters can then be formulated as:

$$\arg\max_{\boldsymbol{\phi}} \mathbb{E}_{p(x,y)} \Big[ \log f(y_i|x_i) \Big].^3 \tag{10}$$

This problem can be solved approximately via gradient-based optimization by leveraging the log-likelihood gradient with respect to parameters $\boldsymbol{\theta}$:

$$\mathbb{E}_{p(x,y)} \Big[ \nabla \log f(y_i|x_i) \Big]. \tag{11}$$

By using estimates of the above expression, denoted $\widehat{\nabla J(\boldsymbol{\theta})}$, we can iteratively refine our distributional parameters $\boldsymbol{\phi}$ via stochastic gradient ascent on $\boldsymbol{\theta}$: $\boldsymbol{\theta} \leftarrow \boldsymbol{\theta} + \alpha \widehat{\nabla J(\boldsymbol{\theta})}$, where $\alpha$ is a positive step-size.

## 3 The principles underpinning scalability in policy gradients

As we discussed in Section 2.2, both Sarsa and Q-learning require maximization of the action-value function: Sarsa relies on greedy action-selection in the limit for optimal convergence and Q-learning needs maximizing the action-value function in the successor state to compute its TD target. Additionally, both Sarsa and Q-learning need action-value maximization in the current state for exploitation or, more generally, for constructing their policies (e.g. an $\varepsilon$-greedy policy relies on choosing greedy actions with probability $1 - \varepsilon$ and uniformly at random otherwise). However, performing exact maximization in complex action spaces is computationally prohibitive. This has in turn limited the applicability of Sarsa and Q-learning to small and finite action spaces. On the other hand, policy gradient methods are widely believed to be suitable for dealing with complex action spaces. In this section, we identify the core principles underlying the scalability of policy gradient methods and describe each such principle in isolation.

### 3.1 Approximate summation or integration using Monte Carlo methods

The scalability of policy gradients in their general stochastic form relies heavily on the identity:

$$\mathbb{E}_{\pi,\mathcal{P}} \Big[ \sum_a Q^\pi(s_t, a) \nabla \pi(a|s_t) \Big] = \mathbb{E}_{\pi,\mathcal{P}} \Big[ Q^\pi(s_t, a_t) \frac{\nabla \pi(a_t|s_t)}{\pi(a_t|s_t)} \Big]$$
$$= \mathbb{E}_{\pi,\mathcal{P}} \Big[ Q^\pi(s_t, a_t) \nabla \log \pi(a_t|s_t) \Big], \tag{12}$$

where the middle expression is derived from our original policy gradient expression (6) by substituting an importance sampling estimator in place of the exact summation or integration over the action space.[4] The rightmost expression is then derived simply by invoking the logarithm differentiation rule, where log denotes the natural logarithm. Consequently, using an experience batch of the usual form $\{(s_t, a_t, r_{t+1}, s_{t+1})\}$ with

---

[3]Equivalent to minimizing the KL-divergence between $p(x, y) = p(y|x)p(x)$ and $\hat{p}(x, y) \doteq f(y|x)p(x)$.

[4]Importance sampling is a Monte Carlo (MC) method used for sampling-based approximation of sums and integrals (Hammersley & Handscomb, 1964).

size $n$, we can construct an estimator of the policy gradient as follows:

$$\frac{1}{n} \sum_t Q^\pi(s_t, a_t) \nabla \log \pi(a_t|s_t), \tag{13}$$

where $Q^\pi$ is the true action-value function under policy $\pi$ which itself needs to be estimated from experience, e.g. via $Q^\pi(s_t, a_t) \approx r_{t+1} + \gamma V(s_{t+1})$ with $V$ serving as a learned approximator of $V^\pi$.

Considering the fact that the policy gradient estimator (13) is founded upon replacing the exact summation or integration over the action space with an on-trajectory (single-action) MC estimator, we can construct a more general class of policy gradient estimators by enabling off-trajectory action samples to also contribute to this numerical computation (Petit et al., 2019):

$$\frac{1}{n} \sum_t \frac{1}{m+1} \Big( Q^\pi(s_t, a_t) \nabla \log \pi(a_t|s_t) + \mathbf{1}_{\{m>0\}} \sum_{i=0}^{m-1} Q^\pi(s_t, a_i) \nabla \log \pi(a_i|s_t) \Big), \tag{14}$$

where $m$ is the number of off-trajectory action samples $a_i \sim \pi(.|s_t)$ per state $s_t$, and the indicator term $\mathbf{1}_{\{m>0\}}$ ensures that when $m = 0$, the second term vanishes, thus reducing to the original on-trajectory policy gradient estimator (13). It is important to note that using the on-trajectory estimator (13) applies broadly to both critic-free (e.g. REINFORCE) and actor-critic (e.g. PPO) methods, but using the off-trajectory estimator (14) with $m > 0$ requires direct approximation of the action-values $Q^\pi$ by a function $Q$, e.g. a neural network trained by minimizing the squared form of the TD residual (3). As such, the latter can only be employed by $Q$-based actor-critic methods (e.g. DDPG).

A large portion of policy gradient algorithms rely on the on-trajectory estimator (13), including REINFORCE (Williams, 1992), A3C (Mnih et al., 2016), and PPO (Schulman et al., 2017b). To our knowledge, surprisingly few algorithms make use of the generalized MC estimator (14), with AAPG (Petit et al., 2019) and MPO (Abdolmaleki et al., 2018) being our only references. On the flip side, methods that perform exact summation or integration over the action space are either limited to small and finite action spaces (Sutton et al., 2001; Allen et al., 2017) or restricted to specific distribution classes that enable closed-form integration (Silver et al., 2014; Ciosek & Whiteson, 2018; 2020).

## 3.2 Amortized maximization using maximum likelihood estimation

In RL and dynamic programming, generalized policy iteration (GPI) (Bertsekas, 2017) represents a class of solution methods for optimizing a policy by alternating between estimating the value function under the current policy (*policy evaluation*) and enhancing the current policy (*policy improvement*). Sarsa is an instance of GPI, wherein the policy evaluation step involves learning of an estimator $Q \approx Q^\pi$ by minimizing the temporal difference (3) and the policy improvement step occurs implicitly by acting semi-greedily with respect to $Q$. Policy gradient methods share a close connection to GPI as well (Schulman et al., 2015). They also alternate between policy evaluation (i.e. estimating $Q \approx Q^\pi$) and policy improvement (i.e. updating an explicit policy using an estimate of the policy gradient). Notably, one can instantiate a policy gradient algorithm by performing the policy evaluation step in the same fashion as Sarsa. From this standpoint, the mechanism employed for policy improvement is the main differentiator between policy gradient methods and action-value methods like Sarsa. In the previous section, we illustrated how policy gradient estimation can be carried out in a computationally scalable manner. In this section, we delve into the question of how updating the policy using policy gradients achieves policy improvement, and how it does so in an efficient manner.

We start with recasting the log-likelihood gradient (11) using RL terminology, replacing the variables $(x, y, i, f)$ with $(s, a, t, \pi)$. Moreover, we reinterpret the expectation computation to be under the joint visitation distribution of state-action pairs within an RL context.[5] Subsequently, we contrast the reframed log-likelihood gradient against the policy gradient (12):

$$\underbrace{\mathbb{E}_{\pi,\mathcal{P}}\Big[ \nabla \log \pi(a_t|s_t) \Big]}_{\text{log-likelihood gradient}} \qquad \text{vs.} \qquad \underbrace{\mathbb{E}_{\pi,\mathcal{P}}\Big[ Q^\pi(s_t, a_t) \nabla \log \pi(a_t|s_t) \Big]}_{\text{policy gradient}}. \tag{15}$$

---

[5]While the reframed log-likelihood gradient is useful for comparison against the policy gradient, it does not per se specify a meaningful optimization problem. This is because the target distribution and its estimator are equivalent (i.e. $\pi(a|s)p(s)$).

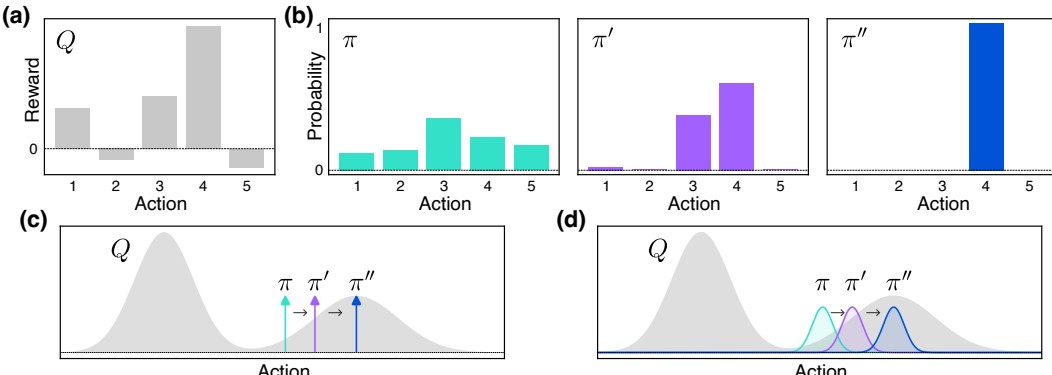

Figure 1: Policy progression according to the true policy gradient in two distinct bandit problems: (a) reward function and (b) softmax-policy progression over time from a random initialization to a deterministic policy in a multi-armed bandit; (c) delta-policy progression in a continuous bandit problem with bimodal rewards; (d) fixed-variance Gaussian-policy progression in the same continuous bandit problem. In (c) and (d), policy progressions overlay the reward function.

This comparison implies that policy gradients perform a modified form of MLE, wherein the log-likelihood gradient term is weighted by $Q^\pi$ for each state-action pair. This weighting assigns importance to actions according to the product of $Q^\pi(s, a)$ and $\log \pi(a|s)$. Therefore, a single step of the true policy gradient updates the policy distribution such that actions with higher action-values become more likely. From this perspective, policy gradients can be construed as a form of amortized inference (Gershman & Goodman, 2014). Each step of the true policy gradient improves the current approximate maximizer of an interdependent action-value function, with the policy functioning as a mechanism for retaining and facilitating retrieval of the best approximation thus far. To elucidate this, we consider a basic one-state MDP (aka. multi-armed bandit) with deterministic rewards (Fig. 1a). In such a setting, true action-values are independent of the policy and are equivalent to rewards: $Q^\pi(a) = Q^*(a) = r(a)$ for all $\pi$ and $a$. For learning, we use a tabular policy function with a softmax distribution and update it using the true policy gradient in each step. These choices minimize confounding effects, allowing us to study the way policy gradients achieve policy improvement in isolation. Figure 1b shows the progression of the policy distribution during training, starting from a random initialization until convergence. Early in training the policy captures the ranking of actions according to their respective action-values. In other words, sampling from the policy corresponds to performing a probabilistic arg sort on the action-value function. In the absence of any counteractive losses, such as entropy regularization, this process continues until convergence to a deterministic policy corresponding to the arg max over the action-value function.

We have discussed that policy gradients can be viewed as an iterative approach to action-value maximization. However, they do not always yield the global arg max. This limitation is rooted in local tendencies of gradient-based optimization, affecting scenarios with non-tabular policy distributions (Tessler et al., 2019). Figures 1c,d respectively show progression of a delta policy and a fixed-variance Gaussian policy in a continuous bandit with bimodal, deterministic rewards. In both cases, policy improvement driven by policy gradients results in local movement in the action space and thus convergence to suboptimal policies.

### 3.3 Representation learning via action-in architectures

There are two functional forms for constructing an approximate action-value predictor $Q$: action-in and action-out architectures. An action-in architecture predicts $Q$-values for a given state-action pair at input. An action-out architecture outputs $Q$ predictions for all possible actions in an input state. Action-out architectures have the computational advantage that a single forward pass through the predictor collects all actions' values in a given state, versus requiring as many forward passes as there are actions in a state by an action-in architecture. Of course, such an advantage is only pertinent when evaluating all possible actions, or a considerable subset of them, in a given state—a necessity that varies depending on the algorithm. On

the other hand, one notable limitation of action-out architectures is their incapacity to predict $Q$-values in continuous action domains without imposing strict modeling constraints on the functional form of the estimated $Q$-function (Gu et al., 2016).

Action-value methods typically use action-out architectures—such as DQN (Mnih et al., 2015) and Rainbow (Hessel et al., 2018)—while policy gradient algorithms with $Q$ approximations rely on action-in architectures to handle complex action spaces—such as DDPG (Lillicrap et al., 2016) and MPO (Abdolmaleki et al., 2018). These choices reflect the distinct requirements of each family. In particular, standard action-value methods must evaluate all possible actions per state to perform maximization, making action-out architectures the more efficient choice from a computational perspective. In contrast, policy gradient methods that rely on $Q$ approximation evaluate only one or a small set of actions in any given state (§3.1). Hence, using action-in architectures in the context of policy gradient methods is more computationally efficient in finite action spaces and one that functionally supports $Q$ evaluation in complex action spaces.

So far, we have compared action-in and action-out architectures from computational and functional standpoints. Now, we turn to a fundamental but often overlooked advantage of action-in architectures: their capacity for representation learning and generalization with respect to actions. Specifically, by treating both states and actions as inputs, action-in architectures unify the process of learning representations for both. For example, when training an action-in $Q$ approximator with deep learning, backpropagation enables learning representations over the joint state-action space. In contrast, action-out architectures are limited in their capacity for generalizing across actions (Zhou et al., 2022). This limitation arises because, although many layers may serve to learn deep representations of input states, action conditioning is introduced only at the output layer in a tabular-like form. While some action-out architectures introduce structural inductive biases that support combinatorial generalization across multi-dimensional actions (see, e.g., Tavakoli et al., 2018; 2021), they do not capacitate action representation learning and generalization in the general form. Moreover, such architectures remain limited to discrete action spaces and are, generally, subject to statistical biases.

## 4 Incorporating the principles into action-value learning

In Section 3, we identified three core principles that we argued underpin the effectiveness of popular policy gradient algorithms in complex action spaces. In this section, we challenge the conventional wisdom that policy gradient methods are inherently more suitable in tackling complex action spaces by showing that the same principles can be integrated into action-value methods, thus enabling them to exhibit similar scaling properties to policy gradient methods without the need for policy gradients.

**Principle 1**   In the same spirit as using an MC estimator in place of exact summation or integration over the action space in policy gradient methods (§3.1), the first principle that we consolidate into action-value learning substitutes exact maximization over the action space with a sampling-based approximation. Formally, we compute an approximation of $\max_a Q(s, a)$ via the steps below:

$$\mathsf{A}_m \doteq \{a_i\}_m \sim \Delta_{\text{search}}(\mathcal{A}_s) \tag{16}$$

$$\arg\max_a Q(s, a) \approx \arg\max_{a_i \in \mathsf{A}_m} Q(s, a_i) \doteq a^{\max} \tag{17}$$

$$\max_a Q(s, a) \approx Q(s, a^{\max}) \tag{18}$$

where $m \geq 1$ is the number of action samples in state $s$ and $\Delta_{\text{search}}$ is a probability distribution over the generally state-conditional action space $\mathcal{A}_s$. Without any prior information, opting for a uniform $\Delta_{\text{search}}$ is ideal as it ensures equal sampling across all possible actions in a given state. This approach, with a constant $m$, allows for action-value learning at a fixed computational cost in arbitrarily complex action spaces. Sampled actions are only used internally to probe the $Q$-function for $\arg\max$ approximation at a given state, and not executed in the environment.

**Principle 2**   The next principle is to equip action-value learning with a mechanism for retention and retrieval of the best $\arg\max$ approximation so far, analogous to the policy function in policy gradient methods (§3.2).

To do so, let us assume we maintain a memory buffer $\mathcal{B} \doteq \{(s_t, a_t^{\max})\}$, where $a_t^{\max}$ denotes our best current $\arg\max$ approximation in a visited state $s_t$. In small and finite state spaces, the memory buffer itself can serve as a basic mechanism for retention and retrieval via table-lookup (as used by Tian et al., 2022):

$$a_t^{\max} \leftarrow \mathcal{B}(s_t) \text{ if } s_t \text{ in } \mathcal{B} \text{ otherwise } \varnothing. \tag{19}$$

In this case, we can enable the reuse of past computations for amortized $\arg\max$ approximations by modifying Eq. 16 in the following way:

$$\mathsf{A}_m \doteq \{a_t^{\max}\} \cup \{a_i\}_{m-1} \sim \Delta_{\text{search}}(\mathcal{A}_s). \tag{20}$$

Then, we refine the $\arg\max$ approximation via Eq. 17 and update the buffer $\mathcal{B}(s_t) \leftarrow a_t^{\max}$. This approach does not achieve generalization across states, thus compromising its general efficacy in major ways. To enable a capacity for generalization, we resort to training a state-conditional parameterized distribution function with MLE (§2.4). In other words, we train a parametric $\arg\max$ predictor $f_{\boldsymbol{\theta}}(.|s_t)$ by employing the log-likelihood gradient (11) on the stored tuples $\{(s_t, a_t^{\max})\}$. Notably, this paradigm naturally supports training an ensemble of such predictors, for example based on different distribution families. Therefore, we can rewrite Eq. 20 to explicitly incorporate an ensemble of $k$ parametric $\arg\max$ predictors as below:

$$\mathsf{A}_m = \bigcup \begin{cases} \mathsf{A}_{m_0} \sim \text{Uniform}(\mathcal{A}_{s_t}) \\ \mathsf{A}_{m_1} \sim f_{\boldsymbol{\theta}_1}(.|s_t) \\ \cdots \\ \mathsf{A}_{m_k} \sim f_{\boldsymbol{\theta}_k}(.|s_t) \\ \{a_t^{\max}\} \text{ (if a prior approximation exists)} \end{cases} \tag{21}$$

**Principle 3** The third, and final, principle is to combine action-value learning with action-in instead of action-out architectures in order to enable action-value inference in complex action spaces as well as representation learning and generalization with respect to actions (§3.3). While the other ingredients apply more broadly to both tabular and approximate cases, this last one is only relevant in conjunction with functional approximation. Appendix B.1 provides a neural network architecture from our experiments that exemplifies the action-in approach.

## 5 Experiments

To evaluate our framework, we instantiate *Q-learning with maximum likelihood estimation* (QMLE) as an example of integrating the adapted core principles (§4) into approximate Q-learning with deep neural networks (Mnih et al., 2015). Appendix A presents the QMLE algorithm in a general form. Our illustrative study (§5.1) employs a simplified implementation of this algorithm. Appendix B provides the details of the QMLE agent used in our benchmarking experiments (§5.2).

### 5.1 Illustrative example

In this study, we present a first illustrative example of how the combination of the three core principles enables QMLE to extend action-value learning to continuous-action problems. Specifically, Principle 1 allows approximate $\arg\max$ computation over non-convex $Q$-surfaces defined on continuous (i.e. non-finite) action spaces, as demonstrated in a 2D action setting. Principle 2 facilitates iterative caching and refinement of the $\arg\max$ approximation, allowing efficient reuse during and after training. Principle 3 enables function approximation of the $Q$-function over continuous actions, which is not possible with action-out architectures such as that used in DQN. Additionally, this example highlights that, with respect to Principles 1 and 2, action-value learning can offer more flexible $\arg\max$ computation than deterministic policy gradients (DPG). As a result, QMLE is able to both mimic and surpass the behavior of DPG, depending on the choice of the action sampling scheme.

We compare QMLE to the deterministic policy gradient (DPG) algorithm in a continuous 2D bandit problem with deterministic and bimodal rewards (similar to that presented by Metz et al., 2019). This problem

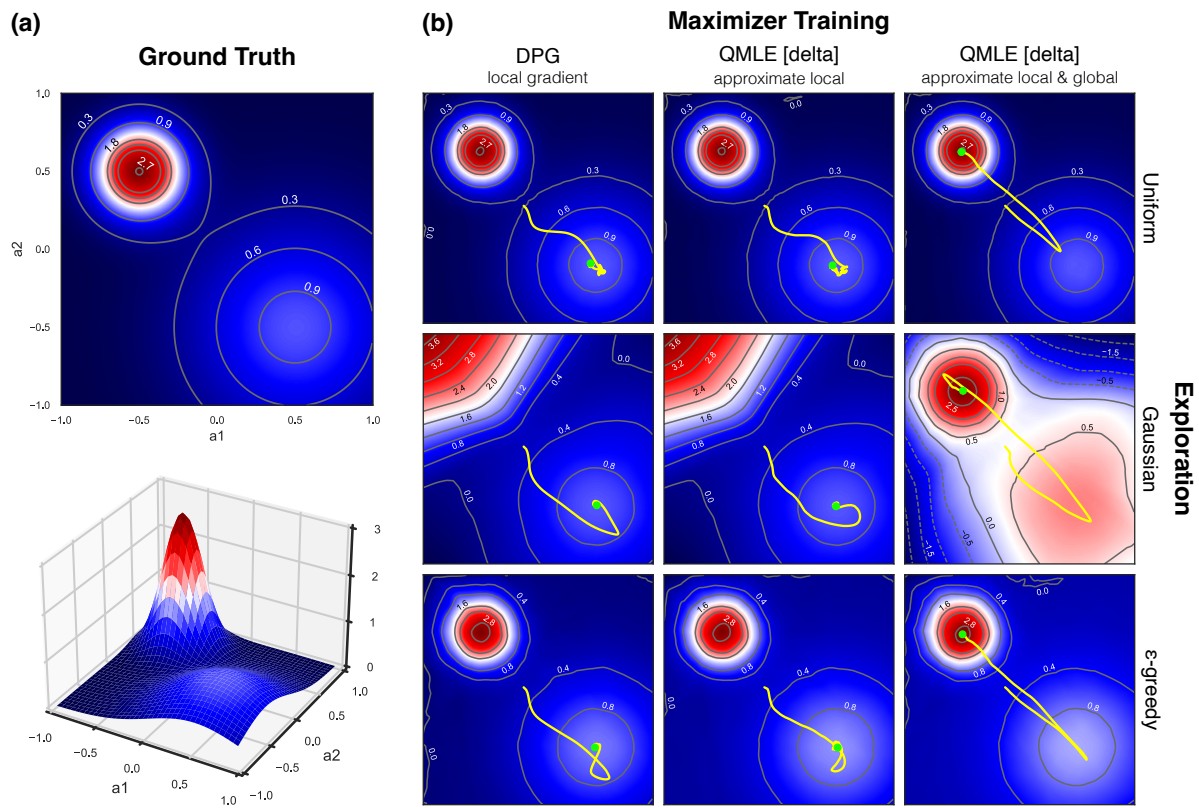

Figure 2: QMLE with local sampling approximately subsumes DPG and with added global sampling transcends DPG by circumventing suboptimality, as examined in a continuous 2D bandit with two modes and under three canonical exploration strategies. The trajectory of delta distributions during training (yellow) with endpoints (green) overlay the respective learned $Q$-functions at convergence.

setting minimizes confounding factors by reducing action-value learning to supervised learning of rewards and eliminating contributions from differing bootstrapping mechanisms in the two methods. For an apples-to-apples comparison, we constrain QMLE to only a single parametric $\arg\max$ predictor based on a delta distribution, mirroring the strict limitation of DPG to delta policies. We further simplify QMLE by aligning its computation of greedy actions with that of DPG. This ensures the only remaining difference between QMLE and DPG is in how their delta parameters are updated, not in how their greedy actions are computed for constructing behavior policies. Both methods use the same hyper-parameters, model architecture, and initialization across all experiments.

We examine two simplified variants of QMLE. The first one samples around the delta parameters for $\arg\max$ approximations that are used as targets for MLE training. Precisely, we only allow samples $\mathsf{A}_m$ drawn from $\delta_{\boldsymbol{\theta}}(s) + \xi$, where $\delta_{\boldsymbol{\theta}}$ denotes the delta-based $\arg\max$ predictor and $\xi$ is a zero-mean Gaussian noise with a standard deviation of 0.001 (cf. Eq. 21). This is akin to computing an MC approximation of $\nabla_a Q^\pi(s_t, a = \pi(s_t))$ in DPG (7c). The second variant additionally incorporates sampling uniformly over the full action space, corresponding to $\mathsf{A}_{m_0}$ samples in Eq. 21. For brevity, we refer to the first variant as using *local sampling*, and the second as using *global sampling*.

Figure 2a depicts the reward function of the bandit, or equally the ground-truth $Q$-function. Figure 2b shows the trajectory of delta distributions during training (yellow) until convergence (green), overlaid on the final learned $Q$-function. DPG (Fig. 2b, left) consistently converges to a local optimum, regardless of the exploration strategy and despite the sufficient accuracy of its learned $Q$-function. QMLE with local sampling (Fig. 2b, middle) behaves similarly to DPG. On the other hand, QMLE with global sampling (Fig. 2b, right) converges to the global optimum across all exploration strategies.

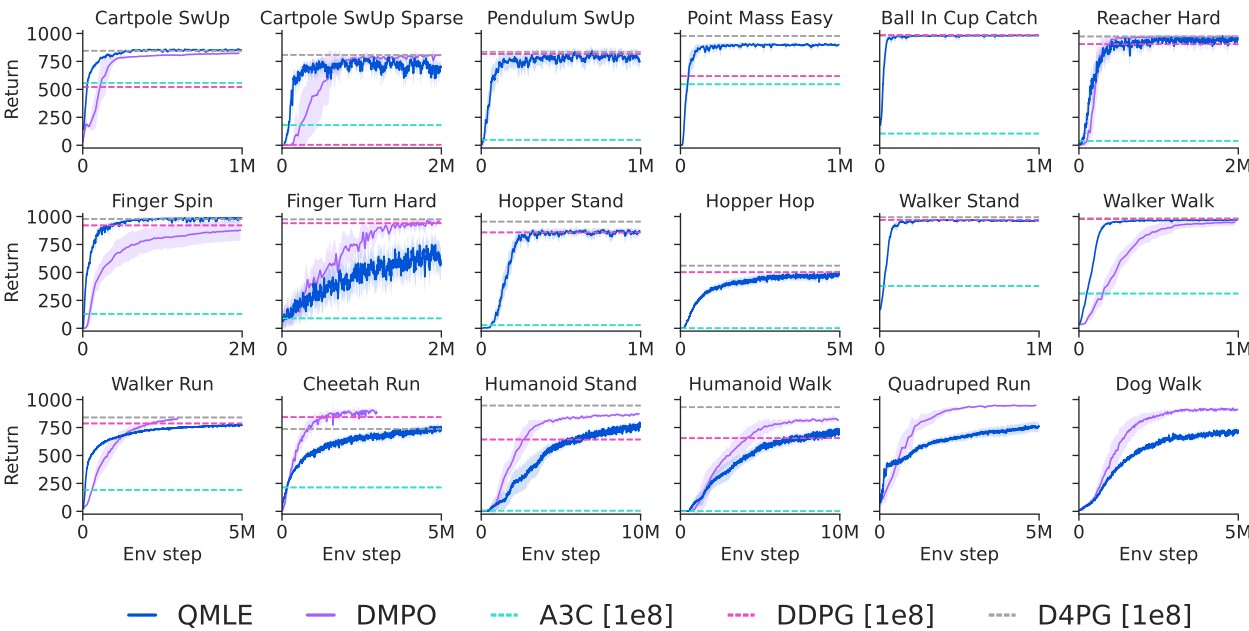

Figure 3: Comparison of QMLE against its closest policy gradient counterpart, off-policy DDPG, with the canonical on-policy A3C included for reference. Performance of state-of-the-art methods DMPO and D4PG is also shown, representing a potential next frontier for QMLE. DMPO and D4PG incorporate advances such as distributional and $N$-step learning, which are not used in our current QMLE implementation. Performance curves (QMLE, DMPO) show mean undiscounted return $\pm$ 1 standard error over seeds. Performance levels (A3C, DDPG, D4PG) report the mean undiscounted return, averaged over 100 evaluation episodes per seed and across seeds. The number of seeds per agent is detailed in Table 2.

This study illustrates key properties of QMLE with respect to DPG: **subsumption**, where QMLE with local sampling approximates DPG updates, and **transcendence**, where global sampling allows QMLE to overcome the local tendencies of policy gradients and surpass DPG.

## 5.2 Benchmarking results

In this section, we evaluate QMLE on 18 continuous control tasks from the DeepMind Control Suite (Tassa et al., 2018). Figure 3 shows learning curves of QMLE alongside the learning curves or final performances of several baselines, including state-of-the-art methods DMPO (Hoffman et al., 2022) and D4PG (Barth-Maron et al., 2018), as well as the canonical (on-policy) A3C (Mnih et al., 2016) and (off-policy) DDPG (Lillicrap et al., 2016). Results for DMPO (12 tasks) are from Seyde et al. (2023), while those for A3C, DDPG, and D4PG (16 tasks) are from Tassa et al. (2018).

With the exception of the *Finger Turn Hard* task, QMLE consistently performs between DDPG and D4PG. Notably, it matches or outperforms DDPG on 14 out of 16 tasks, with DDPG being the closest counterpart from the policy gradient paradigm to QMLE. Moreover, QMLE substantially exceeds the performance of A3C across all tasks. This is despite QMLE being trained on 10 to 100$\times$ fewer steps compared to A3C, DDPG, and D4PG. While QMLE competes well with DMPO in low-dimensional action spaces, it trails in higher-dimensional ones. Nonetheless, the strong performance of QMLE in continuous control tasks with up to 38 action dimensions, all without policy gradients, in and of itself testifies to the core nature of our identified principles and their adaptability to action-value methods.

Appendix C provides supplementary benchmarking results against alternative policy gradient and action-value methods.

## 5.3 Ablation studies

We conducted ablation experiments to assess the contribution of each principle from Section 4 to QMLE's performance and scalability. Here, we provide a concise summary of our findings and refer the reader to Appendix D for the learning curves and additional details.

In Appendix D.1, we study the impact of the number of action samples used for approximating the value-maximizing action. We find that reducing the number of samples from 1000 to as few as 2 has negligible effect on final performance, highlighting a *low sensitivity to sampling budgets*. Although using significantly more samples improves accuracy, the benefits often taper off due to amortized maximization, which enables reuse of previous approximations. Consequently, one can significantly *reduce inference costs* by lowering the sample count without sacrificing performance, a practically valuable property for large-scale tasks or when using large models where each forward pass is computationally expensive.

In Appendix D.2, we ablate amortized maximization by removing parametric $\arg\max$ predictors and relying solely on uniform sampling. Performance degrades significantly under this condition, particularly in higher-dimensional action spaces. This study confirms the critical role of amortization when sampling budgets are limited relative to the action-space complexity, an inevitable problem scenario across real-world domains where action-space complexity quickly outpaces feasible sampling budgets.

Finally, in Appendix D.3, we replace our action-in architecture with an action-out variant. This ablation proves computationally infeasible in large discrete action spaces and underperforms even in moderate action spaces, underscoring the importance of action-in architectures for scalability and generalization.

Taken together, these ablations support that each principle individually contributes to improved scalability or performance of action-value learning.

## 6 Conclusion

In this paper, we distilled the success of policy gradient methods in complex action spaces into three core principles: MC approximation of sums or integrals, amortized maximization using a special form of MLE, and action-in architectures for representation learning and generalization over actions. We then argued that these principles are not exclusive to the policy gradient paradigm and can be adapted to action-value methods. In turn, we presented a framework for incorporating adaptations of these principles into action-value methods. To examine our arguments, we instantiated QMLE by implementing our adapted principles into approximate Q-learning with deep neural networks. Our results showed that QMLE performs strongly in continuous control problems with up to 38 action dimensions, largely outperforming its closest policy gradient counterpart DDPG. These results provided empirical support for the core nature of our identified principles and demonstrated that action-value methods could adopt them to achieve similar qualities, all without policy gradients. In a comparative study using DPG and two simplified QMLE variants, we highlighted a key limitation of policy gradients; namely, their reliance on local updates in continuous parameter spaces, which can lead to convergence to suboptimal solutions in multimodal or non-convex value landscapes. By contrast, QMLE leverages flexible action sampling and explicit maximization to overcome this limitation. This study serves as a motivator for a shift from policy gradients toward action-value methods with our adapted principles. It also offers a potential explanation for the improvements observed over DDPG in our benchmarking experiments.

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

# A    Q-learning with maximum likelihood estimation

In this section, we present the *Q-learning with maximum likelihood estimation* (QMLE) algorithm. Specifically, our presentation is based on integrating our framework (§4) into the deep Q-learning algorithm by Mnih et al. (2015). In line with this, we make use of experience replay and a target network that is only periodically updated with the parameters of the online network. Importantly, we extend the scope of the target network to encompass the arg max predictors in QMLE. Although the algorithm does not mandate the use of action-in $Q$ approximators per se, such architectures become necessary for addressing problems with arbitrarily complex action spaces (§3.3).

Algorithm 1 details the training procedures for QMLE. Notably, the algorithm is flexible regarding the composition of the ensemble of arg max predictors. For instance, the ensemble can consist of a combination of continuous and discrete distributions for problems with continuous action spaces. QMLE introduces several hyper-parameters related to its action-sampling processes. These include the sampling budgets for target maximization, $m_{\text{target}}$, and greedy action selection in the environment, $m_{\text{greedy}}$. Additionally, QMLE uses sample allocation ratios $\{\rho_0, \rho_1, \ldots, \rho_k\}$, where $\rho_0$ corresponds to the proportion of the budget allocated to uniform sampling from the action space, and $\rho_1$ through $\rho_k$ correspond to the proportions assigned to the ensemble of $k$ parametric arg max predictors.

To effectively manage training inference costs in QMLE, we recommend allocating a larger budget to $m_{\text{greedy}}$ than to $m_{\text{target}}$. Since $m_{\text{greedy}}$ is used at most once per interaction step, increasing it incurs relatively little computational burden. In addition, more accurate arg max approximations during training interactions can lead to higher quality data for learning, making this increase particularly beneficial. In contrast, each training update requires $m_{\text{target}} \times N_b$ inferences on the target $Q$-network, where $N_b$ is the batch size. This makes increasing $m_{\text{target}}$ much more costly in terms of training inference costs. On that account, choosing

---

**Algorithm 1:** QMLE algorithm.

---

**Input:** sampling budgets $m_{\text{target}}$, $m_{\text{greedy}}$ and ratios $\{\rho_0, \rho_1, \ldots, \rho_k\}$ ($k$ is the # of arg max predictors)
**Input:** initial model parameters $\boldsymbol{\omega}, \{\boldsymbol{\theta}_1, \boldsymbol{\theta}_2, \ldots, \boldsymbol{\theta}_k\}$; step sizes $\alpha_q, \alpha_{\text{argmax}}$
**Input:** target update frequency $N^-$; batch size $N_b$; replay period $K$; interaction budget $N_e \cdot T$

Initialize target parameters $\boldsymbol{\omega}^-, \{\boldsymbol{\theta}_i^-\}_1^k \leftarrow \boldsymbol{\omega}, \{\boldsymbol{\theta}_i\}_1^k$, accumulators $\Delta_q = \{\Delta_i\}_1^k = 0$
Initialize memory buffer $\mathcal{B} = \varnothing$
**for** *episode* $\in \{1, 2, \ldots, N_e\}$ **do**
    Observe initial state $s_0$
    **for** $t \in \{0, 1, \ldots, T-1\}$ **do**
        **with** *probability* $\varepsilon$ **do**
            Sample action $a_t \sim \text{Uniform}(\mathcal{A}_{s_t})$
        **otherwise do**
            Generate actions $\mathsf{A}_m^{\text{greedy}}$ using $\{\boldsymbol{\theta}_i\}_1^k, \{m_i = \rho_i \times m_{\text{greedy}}\}_0^k$ in Eq. 21,
            Approximate greedy action $a_t$ using $Q_{\boldsymbol{\omega}}, s_t, \mathsf{A}_m^{\text{greedy}}$ in Eq. 17
        Observe $r_{t+1}, s_{t+1}, \gamma_{t+1}$ from environment given $a_t$, set $a_{t+1}^{\max} \leftarrow a_t$
        Store transition $(s_t, a_t, r_{t+1}, s_{t+1}, \gamma_{t+1}, a_{t+1}^{\max})$ in $\mathcal{B}$
        **if** $t \equiv 0 \mod K$ **then**
            **for** $j \in \{1, 2, \ldots, N_b\}$ **do**
                Sample random transition $(s_j, a_j, r_{j+1}, s_{j+1}, \gamma_{j+1}, a_{j+1}^{\max})$ from $\mathcal{B}$
                Generate actions $\mathsf{A}_m^{\text{target}}$ using $\{\boldsymbol{\theta}_i^-\}_1^k, \{m_i = \rho_i \times m_{\text{target}}\}_0^k, a_{j+1}^{\max}$ (prior) in Eq. 21
                Approximate target-maximizing action $a_{j+1}$ using $Q_{\boldsymbol{\omega}^-}, s_{j+1}, \mathsf{A}_m^{\text{target}}$ in Eq. 17
                Set $a_{j+1}^{\max} \leftarrow a_{j+1}$ and update $\mathcal{B}$
                Compute squared TD residual $\mathcal{L}_q = (r_{j+1} + \gamma_{j+1} Q_{\boldsymbol{\omega}^-}(s_{j+1}, a_{j+1}^{\max}) - Q_{\boldsymbol{\omega}}(s_j, a_j))^2$
                Compute MLE losses $\{\mathcal{L}_i\}_1^k$ using parameters $\{\boldsymbol{\theta}_i\}_1^k$ and target $a_{j+1}^{\max}$
                Accumulate parameter-changes $\Delta_q \leftarrow \Delta_q + \nabla_{\boldsymbol{\omega}} \mathcal{L}_q, \{\Delta_i \leftarrow \Delta_i + \nabla_{\boldsymbol{\theta}_i} \mathcal{L}_i\}_1^k$
            Update parameters $\boldsymbol{\omega} \leftarrow \boldsymbol{\omega} + \frac{1}{N_b} \cdot \alpha_q \cdot \Delta_q, \{\boldsymbol{\theta}_i \leftarrow \boldsymbol{\theta}_i + \frac{1}{N_b} \cdot \alpha_{\text{argmax}} \cdot \Delta_i\}_1^k$
            Reset accumulators $\Delta_q = \{\Delta_i\}_1^k = 0$
        Update target parameters $\boldsymbol{\omega}^-, \{\boldsymbol{\theta}_i^-\}_1^k \leftarrow \boldsymbol{\omega}, \{\boldsymbol{\theta}_i\}_1^k$ every $N^-$ time steps
        Terminate episode on reaching a terminal state, where $\gamma_{t+1} = 0$

a moderate $m_{\text{target}}$ allows for computational tractability with larger batch sizes. Remarkably, a moderate $m_{\text{target}}$ could also help reduce the overestimation of action-values (van Hasselt, 2010; van Hasselt et al., 2016). Also, assigning a smaller $m_{\text{target}}$ relative to $m_{\text{greedy}}$ is further justified because target maximization benefits from additional amortization. Specifically, each time a transition is sampled from the memory buffer for experience replay, we use the previously stored arg max approximation as a prior. This approximation is then recalibrated and updated in the memory buffer for the next time that the transition is sampled for replay.

## B  Experimental details

This section details the specific QMLE instance that we evaluated in our benchmarking experiments. We adopted prioritized experience replay (Schaul et al., 2016), in place of the uniform variant that was described in Algorithm 1. Furthermore, we deployed QMLE with two arg max predictors: one based on a delta distribution over the continuous action space, and another based on a factored categorical distribution defined over a finite subset of the original action space (Tang & Agrawal, 2020).

To build the discrete action support, we applied the bang-off-bang (3 bins) discretization scheme to the action space (Seyde et al., 2021). For sampling from the delta-based arg max predictor, we always included the parameter of the delta distribution as the initial sample. Any additional samples were generated through Gaussian perturbations around this parameter using a small standard deviation.

Sections B.1, B.2, and B.3 provide details around the model architecture, hyper-parameters, and implementation of QMLE in our benchmarking experiments, respectively. Section B.4 details the number of seeds per agent and the computation of our learning curves.

### B.1  Model architecture

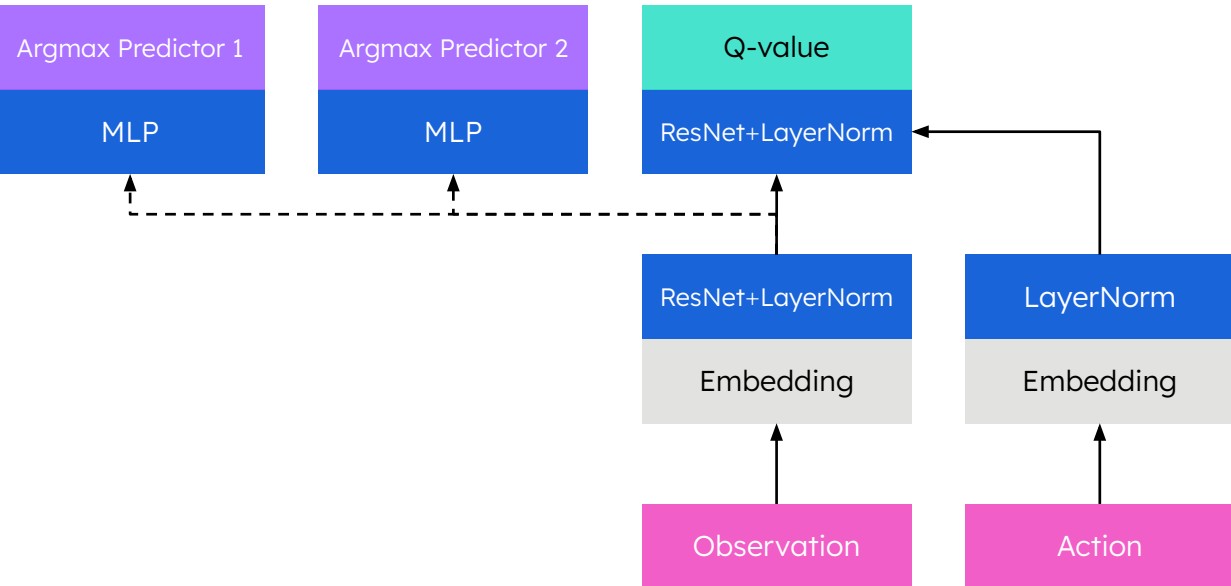

Figure 4: Schematic of the model architecture used with QMLE for our benchmarking experiments. Dashed lines indicate paths without gradient flow during backpropagation.

Figure 4 depicts the model architecture of QMLE in our benchmarking experiments. The model begins with two separate streams, one for the observation inputs and the other for the action inputs. The outputs of these streams are then concatenated and jointly processed by the Q-value predictor. Furthermore, the output of the observation stream is separately processed by each arg max predictor.

In the observation stream, we apply a linear embedding layer with 128 units followed by a residual block (He et al., 2016) that maintains this width and uses rectified linear unit (ReLU) activation (Nair & Hinton, 2010). The residual block is succeeded by a layer normalization (LayerNorm) operation (Ba et al., 2016) and exponential linear unit (ELU) activation (Clevert et al., 2016).

In the action stream, we apply a linear embedding layer with 128 units. The output of the embedding layer is then directly followed by LayerNorm and ELU activation.

The outputs from both streams are concatenated and passed through a joint observation-action residual block with 256 units and ReLU activation. Subsequently, we apply LayerNorm and ELU activation. The outputs are then linearly mapped to a single scalar, representing the predicted $Q$-value.

The output of the observation stream is also used as input to the two arg max predictors. To avoid interference, we prevent backpropagation from the arg max predictor streams through the shared observation stream. Each arg max predictor stream leverages a hidden multilayer perceptron (MLP) layer with 128 units and ReLU activation.

In the arg max predictor stream based on the delta distribution, we produce one output per action dimension. Each output is passed through hyperbolic tangent (Tanh) activation to yield a continuous value constrained within the support of each action dimension in our benchmark. In the arg max predictor stream based on the factored categorical distribution, we produce three outputs per action dimension. We apply the softmax function to the outputs for each action dimension, producing multiple softmax distributions over a bang-off-bang discrete action support.

## B.2    Hyper-parameters

Table 1 provides the hyper-parameters of QMLE in our benchmarking experiments.

Table 1: QMLE hyper-parameters in our benchmarking experiments.

| Parameter | Value |
|---|---|
| $m_{\text{target}}$ | 100 |
| $m_{\text{greedy}}$ | 1000 |
| $\rho_0$ (uniform) | 0.9 |
| $\rho_1$ (delta) | 0.01 |
| $\rho_2$ (factored categorical) | 0.09 |
| step sizes $\alpha_q, \alpha_{\text{arg max}}$ | 0.0005 |
| update frequency | 10 |
| batch size | 256 |
| training start size | 1000 |
| memory buffer size | 1000000 |
| target network update frequency | 2000 |
| loss function | mean-squared error |
| optimizer | Adam (Kingma & Ba, 2015) |
| exploration $\varepsilon$ | 0.1 |
| discount factor | 0.99 |
| time limit | 1000 (Tassa et al., 2018) |
| truncation approach | partial-episode bootstrapping (Pardo et al., 2018) |
| importance sampling exponent | 0.2 |
| priority exponent | 0.6 |

### B.3 Implementation

Our QMLE implementation is based on a DQN script from CleanRL (Huang et al., 2022) and incorporates prioritized experience replay adapted from Stable Baselines (Hill et al., 2018), both available under the permissive MIT license.

To support reproducibility, we release the implementation used in our benchmarking experiments at:
https://github.com/atavakol/qmle

### B.4 Seeds and performance

All curves show the mean undiscounted return across seeds, with one standard error. Performance levels of DDPG, D4PG, and A3C are averaged over 100 episodes per seed, after 100M environment steps of training. Table 2 lists the number of seeds used per agent, grouped by result source.

Table 2: Number of seeds used in benchmarking experiments.

| Agent | Trials |
|-------|--------|
| QMLE | 5  - *Dog* and *Humanoid* tasks |
|       | 10 - all other tasks |
| **Results from** Seyde et al. (2023) | |
| DMPO | 10 |
| DQN | 10 |
| **Results from** Tassa et al. (2018) | |
| A3C | 15 |
| DDPG | 15 |
| D4PG | 5 |
| **Results from** Pardo (2020) | |
| MPO | 10 |
| SAC | 10 |
| TD3 | 10 |
| PPO | 10 |
| TRPO | 10 |
| A2C | 10 |
| **Results from** de Wiele et al. (2020) | |
| AQL | 3 |
| QT-Opt | 3 |

## C Supplementary benchmarking results

Figures 5 and 6 provide comparisons of QMLE with a range of mainstream policy gradient methods. The baseline results are due to Pardo (2020).

- Figure 5 compares QMLE with policy gradient methods that use action-value approximation: MPO (Abdolmaleki et al., 2018), SAC (Haarnoja et al., 2018), and TD3 (Fujimoto et al., 2018).

- Figure 6 compares QMLE with policy gradient methods that use state-value approximation: PPO (Schulman et al., 2017b), TRPO (Schulman et al., 2015), and A2C (Mnih et al., 2016).

Figure 7 compares QMLE with QT-Opt (Kalashnikov et al., 2018) and both the discrete and continuous action variants of AQL (de Wiele et al., 2020). The baseline results are taken from de Wiele et al. (2020).

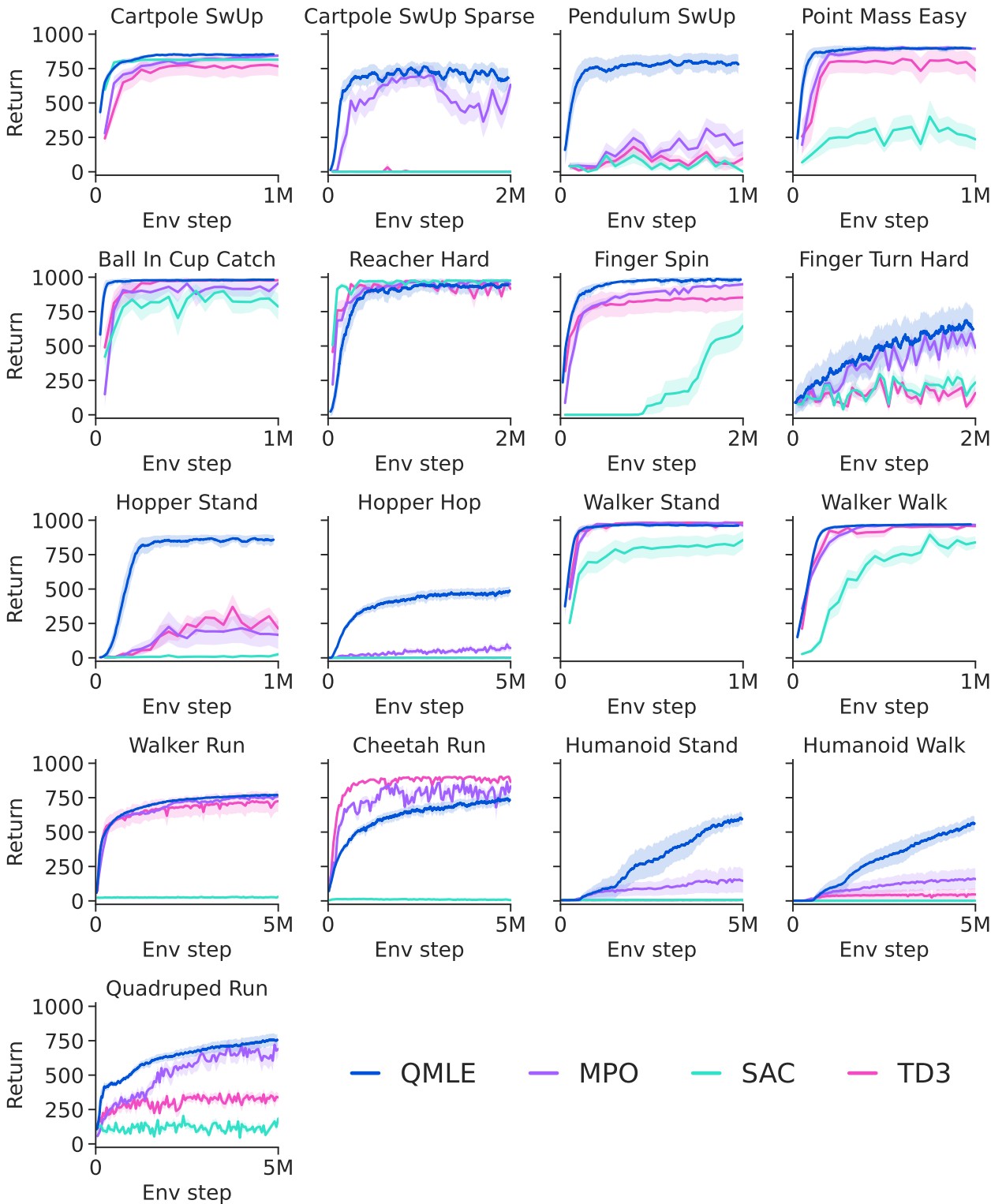

Figure 5: Learning curves of QMLE against MPO, SAC, and TD3.

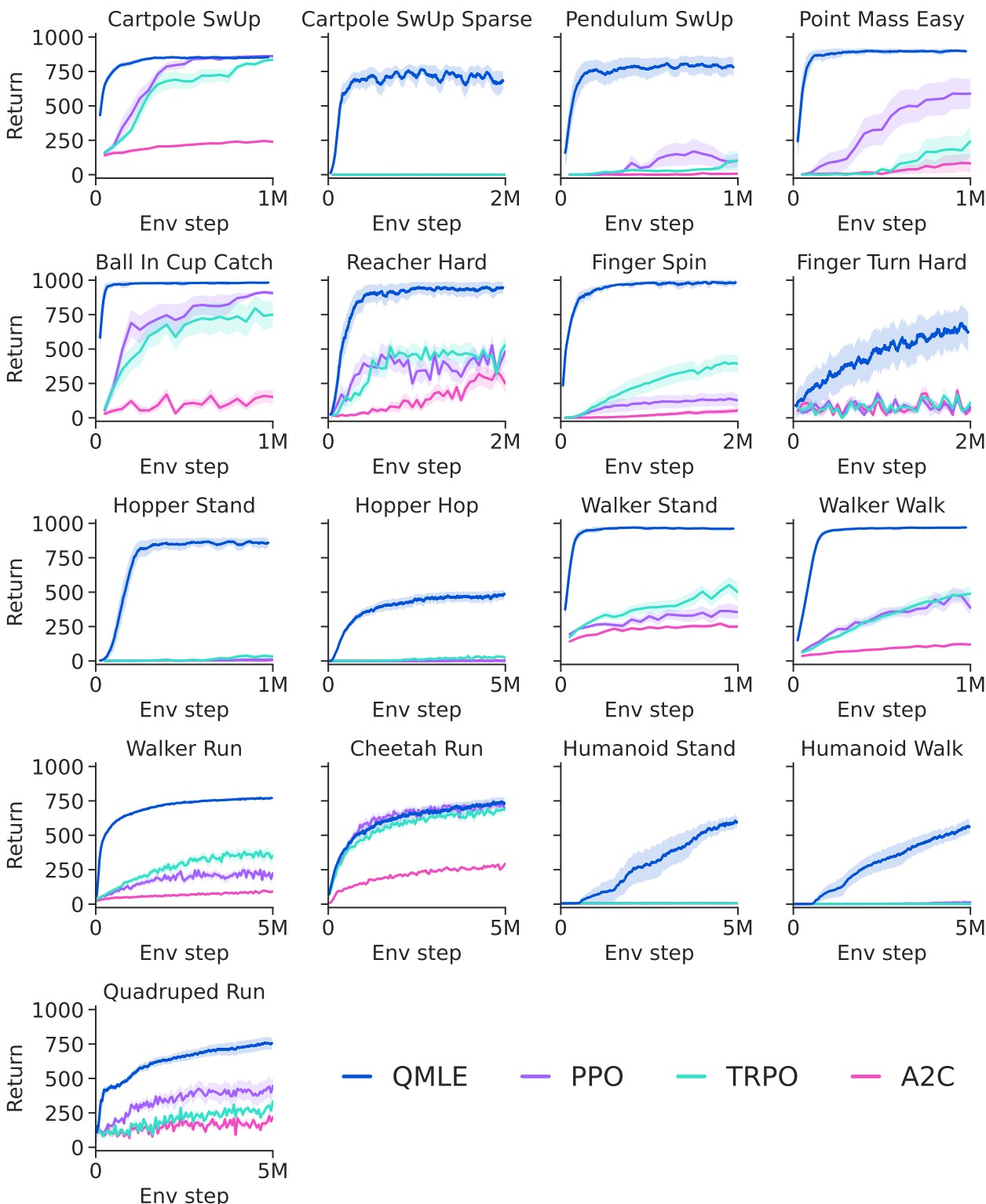

Figure 6: Learning curves of QMLE against PPO, TRPO, and A2C.

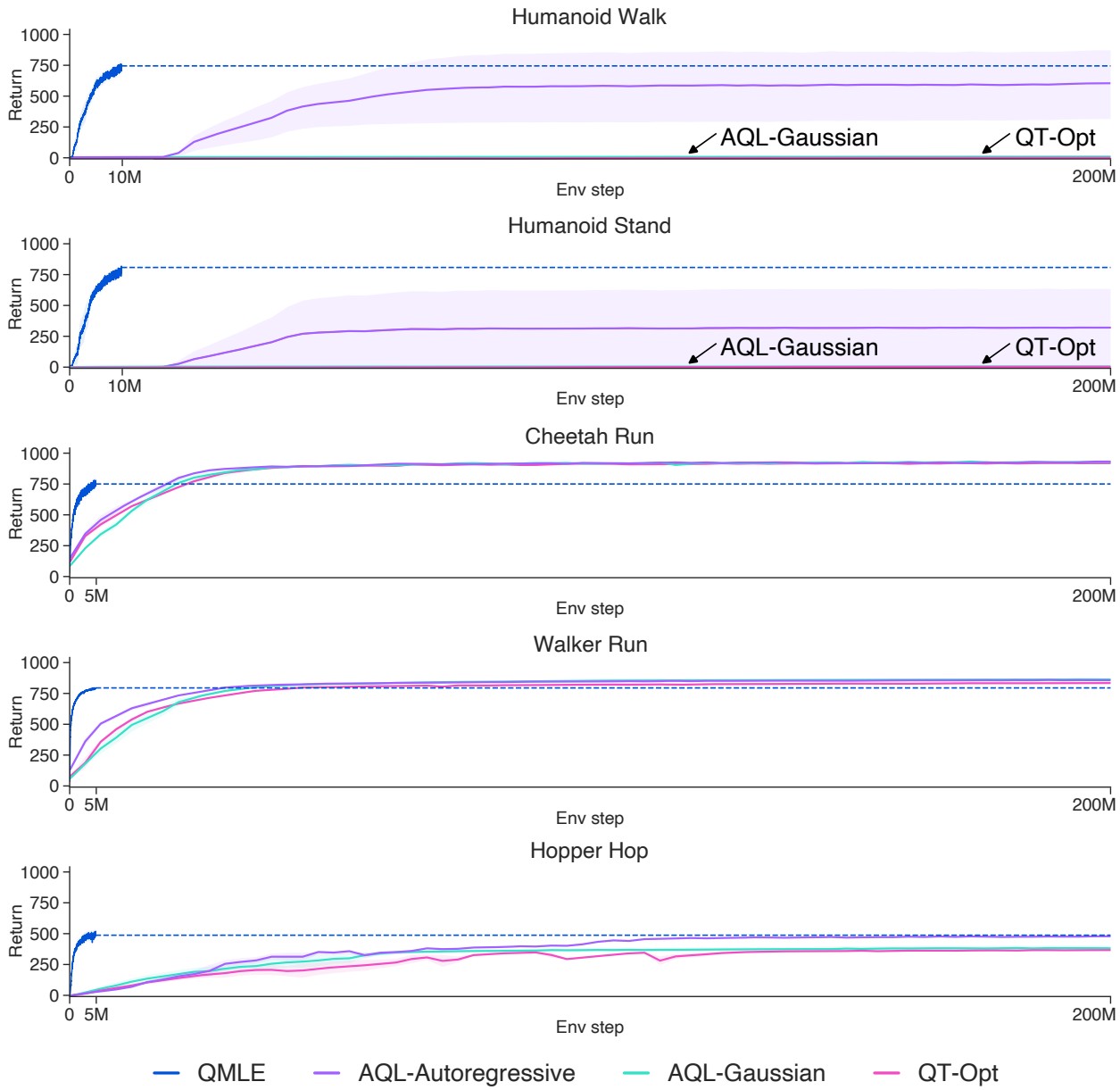

Figure 7: Comparison of QMLE with QT-Opt and AQL.

# D    Ablation studies

In this section, we present ablation studies to evaluate the impact of the principles in our framework on the performance of QMLE.

## D.1    Approximate maximization

Figure 8 shows the learning curves for QMLE with sampling budgets of 2 and 1000. Expectedly, increasing the number of samples for $Q$-maximization improves performance by yielding more accurate estimates of the TD target and greedy actions. Nevertheless, amortization dampens the negative impact of undersampling by enabling reuse of past computations over time.

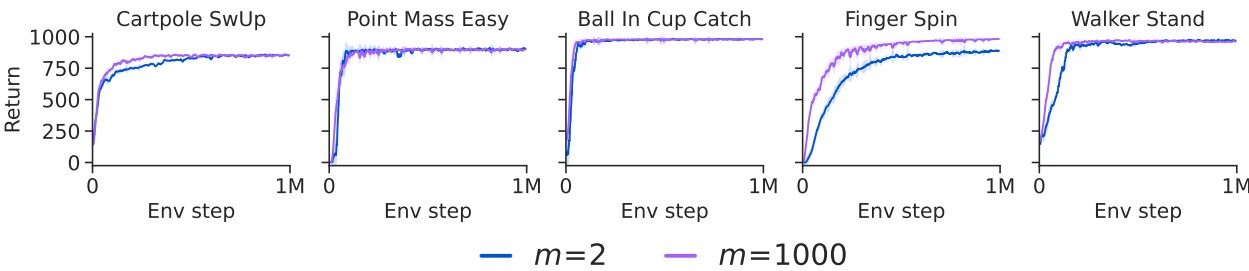

Figure 8:   Comparison of QMLE with sampling budgets of 2 and 1000.

## D.2    Amortized maximization

Figure 9 compares the performance of QMLE against its ablation without amortized maximization. In this experiment, QMLE employs a delta-based $\arg\max$ predictor, while its ablated variant relies solely on uniform sampling for $\arg\max$ approximation. We use the same sampling budgets of $m_{\text{target}} = m_{\text{greedy}} = 2$ for both variants, with QMLE allocating its budgets equally between uniform sampling and the delta-based $\arg\max$ predictor ($\rho_{\text{uniform}} = \rho_{\text{delta}} = 0.5$), and the ablated variant allocating them entirely to uniform sampling ($\rho_{\text{uniform}} = 1$).

The action spaces range from 1-dimensional (leftmost) to 6-dimensional (rightmost) for the considered problems. The results demonstrate that amortized maximization significantly improves performance, particularly as the complexity of the action space increases.

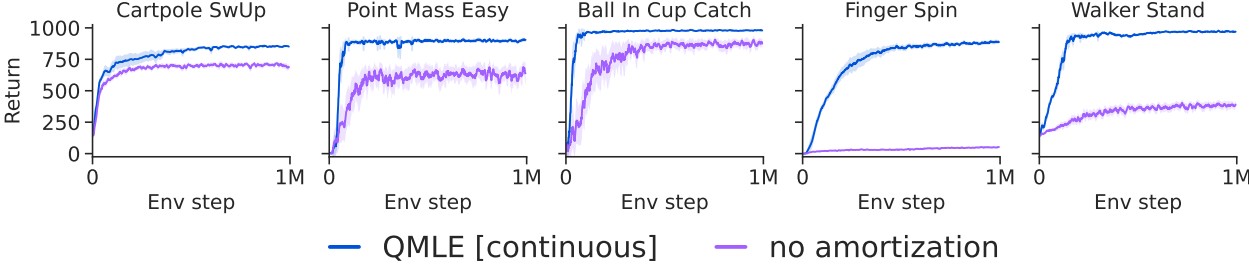

Figure 9:   Comparison of a continuous variant of QMLE with and without amortized maximization.

## D.3    Action-in architecture

We compare the performance of QMLE with action-in and action-out architectures. Since action-out $Q$-approximators are not readily compatible with continuous action spaces, we examine both agents on the bang-off-bang (3 bins) discretized versions of the considered environments.

The QMLE variant with an action-in architecture employs an arg max predictor based on a factored categorical distribution, with the same sampling budgets and uniform sampling ratio as in Table 1 but with $\rho_{\text{delta}} = 0$ and $\rho_{\text{factored categorical}} = 1$. On the other hand, exact maximization is performed for the ablated variant as a forward pass through an action-out $Q$-approximator collects all actions' values in a given state. Therefore, using an action-out architecture in the ablated variant obviates the need for learned arg max predictors or any approximate maximization altogether. That is to say, when inference with an action-out architecture is computationally feasible, performing exact maximization should also be feasible given that its cost is generally negligible compared to that of inference. This, in effect, reduces the ablated variant to DQN.

Figure 10 shows the learning curves for QMLE and DQN.

- In lower-dimensional action spaces, such as *Finger Spin* and *Walker Walk* with 2 and 6 action dimensions respectively, where DQN is computationally tractable, both QMLE and DQN achieve similar final performance levels. However, QMLE performs more sample-efficiently due to the use of an action-in architecture, which enables generalization across actions.

- In higher-dimensional action spaces, DQN becomes computationally intractable, resulting in out-of-memory errors or exceeding computational time constraints. In contrast, QMLE performs strongly in these environments, including *Dog Walk* with $3^{38} \approx 1.35 \times 10^{18}$ discrete actions, underscoring the benefits of action-in architectures both in terms of computational scalability and generalization across enormous action spaces.

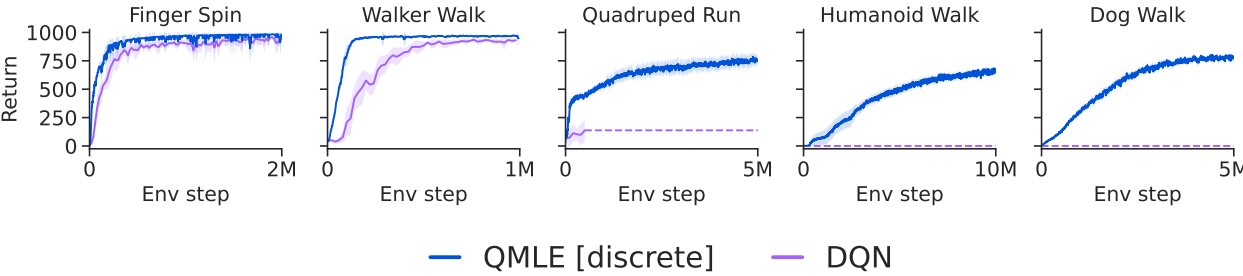

Figure 10: Comparison of QMLE with DQN where DQN represents the ablation of action-in architectures, and in turn all three principles, in QMLE. Dashed lines indicate out-of-memory errors or excessive computational demands for DQN.

# E   Future work

## E.1   Combining with other improvements

In this paper, we integrated our framework into the deep Q-learning algorithm of Mnih et al. (2015), in a proof-of-concept agent that we termed QMLE (Algorithm 1). In our benchmarking experiments, we further combined QMLE with prioritized experience replay (Schaul et al., 2016; see details in Section B). While this setup is relatively basic compared to the advancements in deep Q-learning, it served our purpose of demonstrating the general competency of action-value methods in complex action spaces without involving policy gradients. We anticipate that a purposeful integration with advancements in deep Q-learning could significantly improve the performance of our QMLE agent. For instance, fundamental methods that can be trivially combined with QMLE include $N$-step returns (Sutton & Barto, 2018; Hessel et al., 2018) and distributional learning (Bellemare et al., 2017), similarly to the critics in DMPO and D4PG. Certain methods, including double Q-learning (van Hasselt, 2010; van Hasselt et al., 2016) and dueling networks (Wang et al., 2016) may not be directly applicable or relevant to QMLE, underscoring the importance of careful integration. We are particularly excited about using a cross-entropy classification loss in place of regression for training $Q$ approximators (Farebrother et al., 2024), as well as combining with ideas introduced by Li et al. (2023); Schwarzer et al. (2023). Moreover, formal explorations into the space of value mappings (van Seijen et al.,

2019; Fatemi & Tavakoli, 2022), particularly those that benefit $Q$-function approximation with action-in architectures, offer an intriguing direction for future work.

Since our approach employs maximum likelihood estimation (MLE) in a disentangled manner (see discussions in Section 3.2), it makes it trivial to incorporate advances from supervised learning for training the parametric arg max predictors. To provide an example, advancements in heteroscedastic uncertainty estimation, such that introduced by Seitzer et al. (2022), can be readily applied to model state-conditional variances for Gaussian arg max predictors.

### E.2 Multiagent reinforcement learning via CTDE

A problem scenario that could benefit from QMLE, and more broadly our framework, is multiagent reinforcement learning (MARL) under centralized training with decentralized execution (CTDE; Foerster et al., 2016; Lowe et al., 2017). Currently, the dominant class of solutions in this paradigm is based on combinations of deep Q-learning and value decomposition methods (Sunehag et al., 2017; Rashid et al., 2020). These approaches decompose the $Q$-function into local utilities for each agent, aiming for the local arg max to correspond to the global arg max on the joint $Q$-function. However, maintaining this alignment requires imposing structural constraints that limit the representational capacity of the joint $Q$-approximator, which can lead to suboptimal decentralized arg max policies.

QMLE avoids these constraints by disentangling the process of approximating the joint $Q$-function from learning the decentralized arg max policies, allowing for a universal representational capacity of the joint $Q$-function while maintaining decentralized execution. Instead of relying on a factored $Q$ approximation, QMLE models the joint $Q$-function in an unconstrained manner. Simultaneously, an arg max predictor (or an ensemble of them) is separately trained for each agent, conditioned on their respective observations. This approach allows for improved coordination between agents by preserving the full representational capacity of the joint $Q$-function. As demonstrated in Figure 11, in a continuous variant of the "climbing" game (Claus & Boutilier, 1998), linear value decomposition (Sunehag et al., 2017) leads to a suboptimal reward of 5 due to its constrained capacity to represent the joint $Q$-function as $Q \doteq U1 + U2$. In contrast, QMLE, by accurately modeling the joint $Q$-function, enables decentralized arg max predictors that guide agents to the globally optimal reward of 11.

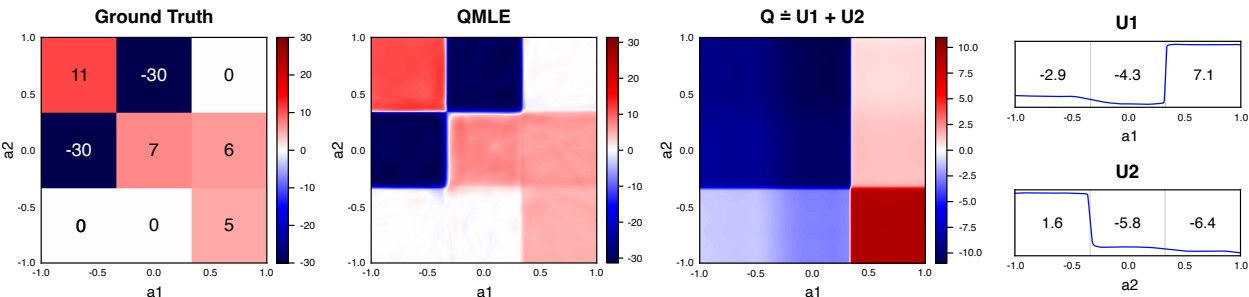

Figure 11: Comparison of QMLE with linear value decomposition in a continuous variant of the "climbing" game with two agents (Claus & Boutilier, 1998). Linear value decomposition leads to a suboptimal reward of 5 due to its limited representational capacity ($Q \doteq U1 + U2$), whereas QMLE, by modeling the joint $Q$-function without such constraints, enables decentralized arg max predictors that guide the agents to the globally optimal reward of 11.

### E.3 Curriculum shaping through growing action spaces

Growing of the action space as a form of curriculum shaping is an effective approach for improving learning performance in complex problems. Nonetheless, existing approaches, such as that presented by Farquhar et al. (2020), are restricted to discrete actions. Seyde et al. (2024) report improvements in sample efficiency and solution smoothness on physical control tasks by adaptively increasing the granularity of discretization

during training. This is because coarse action discretizations can provide exploration benefits and yield lower variance updates early in training, while finer control resolutions later on help reduce bias at convergence. However, due to the strict dependence of this approach on a class of action-out architectures (Tavakoli et al., 2021; Seyde et al., 2023), it cannot ultimately transition from coarse discretization to the original continuous action space.

On the other hand, QMLE can support learning with dynamically growing action spaces, including transitions from finite to continuous supports in continuous action problems. We show this capability in a preliminary experiment, where we start with a coarse bang-off-bang discretization and later shift to the original continuous action space (Figure 12). This capacity positions QMLE, and more broadly our framework, as a promising candidate for future research in the context of growing action spaces.

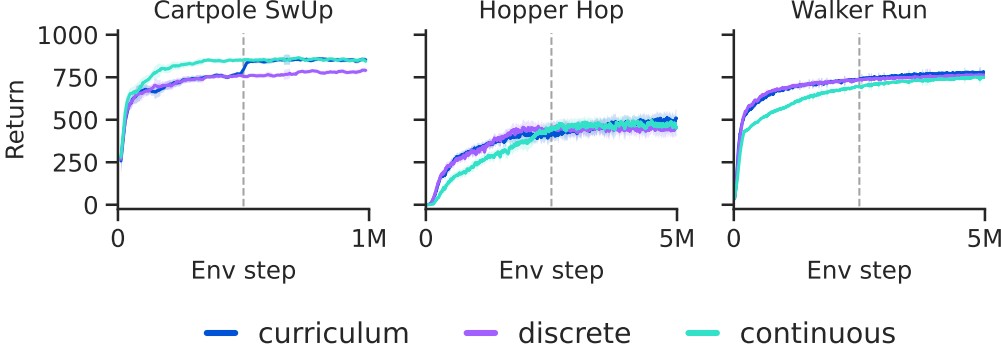

Figure 12: Learning curves for discrete, continuous, and discrete-to-continuous ("curriculum") variants of QMLE. Dashed lines mark the transition from discrete to continuous actions for the curriculum-based agents.