# OpenReview forum: "Learning in complex action spaces without policy gradients"
_TMLR — Accepted by TMLR_

### Review · Reviewer_2ZYF · 2025-02-19

**Summary Of Contributions:**

This paper addresses the following question: why do the computational applicability and performance of policy gradient (PG) methods and action-value (AV) methods diverge as the complexity of the action space increases?

The authors identify three principles that underpin the scalability of PG methods and hypothesize that AV methods can achieve comparable efficiency by incorporating these principles. To validate this hypothesis, they introduce QMLE, an AV method that integrates the three principles, and evaluate it through experiments on a 2D continuous bandit problem and the DeepMind Control Suite. The results demonstrate that QMLE performs on par with popular PG methods, concluding that AV methods combined with the three principles work.

**Audience:**

Yes

**Claims And Evidence:**

No

**Requested Changes:**

1. **Clarify "complex" action space**:
   - If "complex" refers to hybrid action spaces, please include experiments in environments with hybrid action spaces to demonstrate that QMLE performs on par with PG methods.  Also, it is necessary to cite papers discussing RL for hybrid action spaces, such as P-DQN and HyAR, and position this paper into an appropriate context.
   - If "complex" refers to higher-dimensional continuous action spaces, please explicitly define what constitutes the "complexity" of an action space.

2. **Clarify AV methods**:
   - Does "AV methods" refer to algorithms that do not use policy gradients? If so, how do other policy search algorithms fit into this definition?
   - Alternatively, does "AV methods" refer to algorithms that do not use a policy network? If so, how do classification-based policy iteration algorithms fit into this framework?

3. **Improve conciseness of Section 3**:
   - Section 3 could be significantly condensed, as many of the stated principles appear to reiterate well-known facts about PG and reinforcement learning. For example, the equivalence between reward-weighted maximum likelihood estimation and PG has been established in prior work such as [Kober and Peters 2009](https://papers.nips.cc/paper_files/paper/2008/hash/7647966b7343c29048673252e490f736-Abstract.html).

Without clarifying 1 and 2, it is impossible to evaluate the soundness of the paper, which is the most important acceptance criteria in TMLR. After aforementioned revisions and confirming all claims are supported by evidences, I am inclined to acceptance.

**Strengths And Weaknesses:**

**Strength**

- The paper is well-written and easy to follow, with no significant technical inaccuracies aside from minor details.

**Weakness**

- The paper is somewhat lengthy and could benefit from conciseness.
- The central claim is ambiguous. Specifically, it is unclear what is meant by a "complex" action space and AV methods.
- The paper's scope may not appeal to a broad audience.

---

> ### Author Response · Authors · 2025-03-27
> **Response to Reviewer 2ZYF**
>
> We thank the reviewer for the thoughtful feedback and insightful questions. Below, we respond point by point.
>
> ---
>
> ### **Q: Clarify what you mean by “complex” action spaces**
>
> Thank you—this was indeed not clearly defined originally. We now explicitly define “complex action spaces” in the Introduction, referencing Hubert et al. (2021), as:
>
> > “...action spaces for which enumerating actions or brute-force operations are computationally intractable, such as domains with multi-dimensional continuous or high-dimensional discrete actions.”
>
> ---
>
> ### **Q: If "complex" refers to higher-dimensional continuous action spaces, please explicitly define what constitutes the "complexity" of an action space.**
>
> We now address this in the revised Introduction. Specifically:
>
> - Increasing **action dimensionality** causes exponential growth in the action space, rendering brute-force operations (e.g., summation, integration, maximization) computationally intractable.
> - **Continuity** introduces an orthogonal axis of complexity, as continuous action spaces cannot be enumerated at all.
>
> We point readers to the relevant experiments and appendix sections that test QMLE's scalability in:
>
> - High-dimensional continuous action spaces (up to 38D)
> - High-dimensional discrete action spaces (e.g., $3^{38} \approx 1.35 \times 10^{18}$ discrete actions)
>
> ---
>
>
> ### **Q: Clarify what you mean by action-value (AV) methods**
>
> We acknowledge this important distinction and have clarified it in the revised manuscript.
>
> In our work, we use “action-value methods” primarily to refer to variants of **Q-learning and Sarsa**, which optimize $Q(s, a)$ directly—without relying on an explicit policy network or policy gradient updates. We now clarify this scope in the Introduction to avoid implying generality over all possible forms of $Q$-based methods. We also appreciate the pointer to classification-based policy iteration, and will consider integrating this perspective in future iterations.
>
> ---
>
> ### **Q: Section 3 could be more concise**
>
> Thank you for the feedback. We agree that some of the conceptual connections in Section 3, such as the link between policy gradients and maximum likelihood estimation, have been noted in prior literature (e.g., in the context of reward-weighted regression). However, these connections are often presented within a different framing.
>
> Our goal in Section 3 is not to claim novelty for individual insights, but to synthesize and reframe them in a unified structure that highlights their relevance to action-value methods—which is a less commonly explored direction. In particular, we aim to make the cross-paradigm parallels actionable by articulating how each principle contributes to computational scalability and performance in complex action spaces.
>
> We recognize that some readers may already be familiar with portions of this exposition, while others may benefit from the step-by-step treatment—especially those less versed in both paradigms. Additionally, reviewer 9Qh3 noted that they enjoyed the exposition, so we have opted to retain it in the current form, while continuing to look for opportunities to tighten phrasing without sacrificing clarity.

---

> > ### Comment · Reviewer_2ZYF · 2025-04-03
> >
> > Thank you very much. I confirmed all claims are supported by evidences and recommended acceptance.

---

> > > ### Author Response · Authors · 2025-04-14
> > > **Thank You for Your Review**
> > >
> > > Thank you for your thorough review and for your positive recommendation. We appreciate your thoughtful feedback and the time you took to evaluate our work.

---

### Review · Reviewer_9Qh3 · 2025-02-23

**Summary Of Contributions:**

This paper identifies three core principles underlying the success of policy gradient methods and demonstrates that these principles can be adapted to action-value methods. The principles are: (1) Monte Carlo approximation for integration, (2) amortized maximization through maximum likelihood estimation, and (3) action-in architectures for state-action representation learning. The paper introduces QMLE, an action-value method that uses
1. A sampling-based MC approximation of argmax Q(s,a)
2. When sampling actions to approximate argmax Q(s,a), sample actions from a family of distributions that approximate current argmax Q(s,a). These functions are analogous to the agent's policy in policy gradient learning.
3. An action-in architecture to compute Q(s,a) to enable state-action representation learning.

QMLE matches or outperforms policy gradient methods like DDPG on many DeepMind Control Suite tasks, supporting the claim that action-value methods can be effective in complex action spaces when incorporating these universal principles.

**Audience:**

Yes

**Broader Impact Concerns:**

None.

**Claims And Evidence:**

Yes

**Requested Changes:**

1. The ablations in Appendix D are critical to the core claim of the paper -- namely, that we can improve action-value algorithms using each of the three principles outlined in the paper -- and should be moved to the main paper. The main experiments in section 5 show that all three principles together can improve performance. If you keep these experiments in the appendix, the main paper should call out their existence and summarize their findings.

1. Clarification question: In Section 4, actions sampled from $A_m$ are only used to probe the Q function for the argmax, correct? These actions are never executed in the environment, correct?

2. The illustrative experiment in Figure 2 was a bit unclear to me at first. This is the first time the term "global sampling" appears in the paper, though it's unclear was this refers to. My initial impression was that QMLE was finding the optimal solution because it was sampling off-policy, which allowed it to explore actions that DPG wasn't explore. After some thought, I think the core takeaway of this experiment is that QMLE is doing a better job approximating argmax Q(s,a) than DPG and that it is important to sample off-policy data *to probe Q(s,a) for the maximizing action*. I think that's the point that needs to be better emphasized here: DPG has a good enough approximation of Q*, but it's not finding a*. If this is the correct interpretation, it's worth emphasizing that this example focuses on principles 1 and 2.

3. Also does the true policy gradient point towards the optimal policy in the illustrative example?

The changes below are more minor:

4. Section 5: "with DDPG being the closest counterpart from the policy gradient paradigm to QMLE." This should be emphasized earlier in the experiments.

5. Conclusion: "we highlighted a key limitation of policy gradients and showed how QMLE could overcome it." Specify the key limitation.

6. The experiment section or figure captions should specify what is plotted (mean with +/- 1 standard deviation). This information is currently in the appendix.

**Strengths And Weaknesses:**

# Strengths
1. Overall, the paper is well-written. I enjoyed the exposition in sections 1-3.
2. The topic of the paper is important, and a portion of the RL community would benefit from the ideas in this paper. I personally would share the paper with colleagues.

# Weaknesses

The paper aims to understand the core principles that bridge the gap between policy gradient algorithms and action-value algorithms, though this goal seems vague. The paper does a good job contrasting these two algorithms, though it remains unclear what it means to successfully "bridge the gap." My interpretation is that (1) each of the three principles of policy gradient algorithms should *individually* improve the data efficiency of action-value learning, and (2) using *all three* principles in action-values algorithms should improve data efficiency more than using a subset of these principles. The experiments in the main paper explore (1), and the ablations in Appendix D explore (2). Since QMLE builds off of Q-learning, DDPG is the closest continuous-action counterpart (the paper mentions this later), QMLE vs DDPG is the comparison we care about most. The paper should more clearly articulate *what* the goal is and *how* the experiments demonstrate that QMLE achieves these goals.

---

> ### Author Response · Authors · 2025-03-27
> **Response to Reviewer 9Qh3**
>
> We thank the reviewer for the encouraging feedback and constructive suggestions. Below we respond point by point.
>
> ---
>
> ### **Q: Clarify what it means to “bridge the gap” between PG and AV methods**
>
> Agreed—this required clearer articulation. In the revised Introduction, we now clarify that:
>
> > “Bridging the gap” refers to identifying and adapting general-purpose principles—traditionally used in policy gradient (PG) methods—for use in action-value methods, so as to enable similar scalability and performance in complex action spaces.
>
> We now state upfront that QMLE vs. DDPG is the central comparison, given that DDPG is QMLE’s closest PG counterpart. This is also emphasized again in Section 5 (Experiments).
>
> ---
>
> ### **Q: Clearly articulate what the goal is and how the experiments demonstrate that QMLE achieves them**
>
> We agree. We now clearly state in Introduction that:
>
> - Ablation studies provide support for the individual contribution of each principle to scalability or performance.
>
> - The three principles collectively enable QMLE to achieve competitive performance against its closest PG counterpart, DDPG.
>
> ---
>
> ### **Q: The ablations in Appendix D are critical—highlight more prominently**
>
> We agree. We now include a subsection in Section 5 summarizing the three main ablation experiments:
>
> 1. Sampling budget sensitivity (Principle 1),
> 2. Effect of amortization (Principle 2),
> 3. Importance of action-in architecture (Principle 3).
>
> Each is briefly explained, and we refer readers to Appendix D for full details and learning curves. This makes it clear that each principle independently contributes to scalability and/or performance.
>
> ---
>
> ### **Q: Clarify whether sampled actions (Section 4) are used for exploration or for computing argmax**
>
> We have clarified in Section 4 (end of Principle 1) that:
>
> > “Sampled actions are only used internally to probe the $Q$-function for argmax approximation at a given state, and not executed in the environment.”
>
> ---
>
> ### **Q: Clarify the goal of the Illustrative example and the term “global sampling”**
>
> Thank you for raising this—your interpretation is largely correct. The illustrative experiment is primarily intended to demonstrate how QMLE leverages action sampling (Principles 1 and 2) to approximate and refine argmax$~Q(s,a)$, even in a continuous, non-convex action space. One important addition, however, is that QMLE’s use of an **action-in architecture** (Principle 3) is what enables it to learn a $Q$-function over continuous actions—something not feasible with action-out models such as DQN. This makes the example our first demonstration of QMLE operating effectively in a **continuous-action domain**, combining all three principles.
>
> In the revised section, we clarify that:
>
> - “Global sampling” refers to drawing actions uniformly over the entire action space (as opposed to local sampling around the delta, as done in DPG).
>
> - This broader sampling enables QMLE to escape local optima that DPG cannot.
>
> - The illustrative task demonstrates the full integration of all three principles and serves as a minimal, clean example of action-value learning in continuous action spaces.
>
> ---
>
> ### **Q: “Also does the true policy gradient point towards the optimal policy in the illustrative example?”**
>
> We assume this refers to the continuous bandit used in the illustrative example (Fig. 2). Please let us know if you had a different scenario in mind.
>
> In that setting, QMLE with purely local sampling mimics DPG by updating the delta policy along the locally increasing direction of the $Q$-surface. As such, the direction of the true policy gradient depends on initialization—it may point toward either mode in the bimodal reward landscape. In general, the true policy gradient does **not** point to the globally optimal solution unless the local gradient happens to be aligned with it. This illustrates a known limitation of policy gradients in multimodal, non-convex settings with continuous actions.
>
> ---
>
> ### **Minor points:**
>
> - We now highlight DDPG as the main PG baseline in Sections 1 and 5.
>
> - In the Conclusion, we specify the key limitation of policy gradients as their **local** update nature (in continuous parameterizations), which can prevent global optimality.
>
> - Figure 3’s caption now specifies that we report mean undiscounted return +/- 1 standard error across seeds. For the appendix figures, we did not repeat this information in every caption, as Appendix B.4 explains the evaluation setup and performance metrics used throughout. However, please let us know if you feel it would be helpful to reiterate this detail in the supplementary captions as well.

---

### Review · Reviewer_rMvR · 2025-03-17

**Summary Of Contributions:**

In RL problems with large, complex action spaces, this manuscript identifies three principles that enable the practical computational advantages of PG methods and applies them to action-value methods, making them more scalable.

**Audience:**

Yes

**Broader Impact Concerns:**

NA.

**Claims And Evidence:**

No

**Requested Changes:**

- What does "complex" action spaces mean? How are you defining the complexity of action space?
- Page 5 Eqn 14: the indices related to $m$ appear to be inconsistent, specifically $\frac{1}{m+1}$ and $\sum_{i=0}^{m-1}$
- Page 6 Sec 3.2: the PG as presented in the paper seems to be actor-critic methods as it also includes the policy evaluation step. Pure PG does not have a value-based component for policy evaluation. In fact, it is concerning that "actor-critic methods" are not mentioned at all in the paper main text, given this paper studies the relationship between value-based and policy-based RL methods.
- Page 6 Eqn 5: why do we want to maximize the policy log likelihood?

**Strengths And Weaknesses:**

- Organization: the main text is short, and a lot of important content and experiments are in the appendix.
- Experimental results are not convincing: in Fig 3, proposed QMLE seems to underperform DMPO in many cases.
- Missing computational scalability results: one of the main claims is the three principles identified are reasons for PG's scalability wrt action space complexity, and those principles can be applied to action-value methods too. However, there are no experiments showing as action space complexity grows, what happens to the performance of different approaches.

---

> ### Author Response · Authors · 2025-03-27
> **Response to Reviewer rMvR (1/2)**
>
> We thank the reviewer for the constructive feedback and helpful suggestions. Below, we address each of the comments raised.
>
> ---
>
> ### **Q: What does "complex" action spaces mean?**
>
> We agree that this term requires a clearer definition. In the revised manuscript, we now explicitly define “complex action spaces” in the Introduction, referencing Hubert et al. (2021), as:
>
> > “...action spaces for which enumerating actions or brute-force operations are computationally intractable, such as domains with multi-dimensional continuous or high-dimensional discrete actions.”
>
> We also now clarify in the Introduction how our experiments feature a diverse range of action complexities. Briefly, we evaluated performances across **1-38 action dimensions** in both **continuous** and **discretized** action spaces. In the discrete-action case, we have shown the effectiveness of QMLE in action-space sizes from 9 to 1.35 quintillion discrete actions.
>
> ---
>
> ### **Q: Equation 14 – Indexing inconsistency ($\frac{1}{m+1}$ vs. $\sum_{i=0}^{m-1}$)**
>
> The division by $m+1$ is due to having 1 on-trajectory sample (left term) plus $m$ off-trajectory samples ($\sum_{i=0}^{m-1}$). Nevertheless, as you noted, there is an inconsistency in the edge case of $m=0$ which results in 1 instead of 0 off-trajectory samples. We have now incorporated an indicator function to handle this case, ensuring mathematical correctness in all cases. This correction appears in the revised Equation 14. Thank you for catching this.
>
> ---
>
> ### **Q: Clarify that policy gradients (PG) includes actor-critic methods**
>
> We appreciate this important point. Throughout the paper, we refer to “policy gradient methods” as encompassing all algorithms derived from the policy gradient theorem (Sutton et al., 1999), which includes both critic-free (e.g., REINFORCE) and actor-critic methods (e.g., DDPG, PPO, TRPO). In summary:
>
> #### **Section 3.1 (Monte Carlo approximations):**
>
> - Equations 12 and 13 apply broadly to all PG methods (critic-free and actor-critic methods).
>
> - Equation 14 applies only to Q-based actor-critic methods (e.g. DDPG, TD3, MPO).
>
> #### **Section 3.2 (Amortized maximization via MLE):**
>
> - Arguments apply broadly to all PG methods (critic-free and actor-critic methods). The policy evaluation step can be executed using MC rollouts (as in REINFORCE), inference on a learned critic (as in DDPG), or a mixture of MC rollouts and critic inference (as in GAE used by PPO).
>
> #### **Section 3.3 (Action-in architectures):**
>
> - Applies primarily to PG methods that learn Q (i.e., actor-critic methods like DDPG, DMPO).
>
> However, we acknowledge that our exposition was not fully clear in the original version. To clarify, we have made the following changes in the revised version:
>
> - We now explicitly state in the Introduction that actor-critic methods are considered variants of policy gradient methods.
>
> - In Section 3.1, we improved clarification around the applicability of on-trajectory and off-trajectory estimators.
>
> ---
>
> ### **Q: Equation 5 – Why maximize policy log-likelihood?**
>
> We believe the reviewer meant Equation 15. (Please let us know if you were referring to a different expression.)
>
> In Eq. 15, we are not suggesting to maximize the log-likelihood directly, but instead use it to illustrate a conceptual parallel between MLE and policy gradients. We contrast the standard log-likelihood gradient with the reward-weighted log-likelihood in PG to show how PG can be seen as a modified MLE.
>
> To clarify this further, we have added a note in Footnote 5 to explicitly explain that this is not an optimization objective, but a reframing to facilitate conceptual understanding.
>
> ---
>
> ### **Q: Organization – Too much in appendix; main text is short**
>
> We appreciate this suggestion. In the revised version:
>
> - We have added a new subsection in Section 5 (Experiments) to summarize the key findings of our ablation studies from Appendix D.
>
> - This summary explicitly shows the effect of each principle on performance and scalability.
>
> This change ensures that readers can grasp the importance of each principle without needing to leave the main body.

---

> > ### Author Response · Authors · 2025-03-27
> > **Response to Reviewer rMvR (2/2)**
> >
> > ### **Q: Experimental results – QMLE underperforms DMPO in many cases**
> >
> > Thank you for pointing this out. We have clarified in Introduction and Section 5 that:
> >
> > - Our primary comparison target is DDPG, as it is the most directly comparable policy-gradient counterpart to QMLE in the continuous action space setting.
> >
> > - DMPO and D4PG are included only as references for future directions and represent state-of-the-art systems that incorporate additional techniques such as distributional and N-step learning—which our implementation of QMLE does not include.
> >
> > Additionally, we would like to highlight that:
> >
> > - In Appendix E, we show that QMLE significantly outperforms other common baselines such as MPO, SAC, TD3, PPO, A2C, AQL, and QT-Opt.
> >
> > - As stated in the Introduction, the goal of our paper is not to claim SOTA performance, but to demonstrate that action-value methods—when equipped with these three principles—can perform competitively in complex action spaces, even without using policy gradients.
> >
> > ---
> >
> > ### **Q: No experiments showing as action space complexity grows, what happens to the performance of different approaches.**
> >
> > We agree this is important to highlight. While we did not include runtime or FLOPs benchmarks, we evaluated QMLE—alongside several baselines—across a broad spectrum of action dimensionalities, ranging from 1D (CartPole SwingUp) up to 38D (Dog Walk). This exponential growth in action space size renders brute-force operations such as summation, integration, or maximization increasingly intractable.
> >
> > Moreover, continuity of the action space introduces a separate axis of complexity, further preventing enumeration-based strategies. Despite these challenges, QMLE maintains strong performance across both continuous and discretized action spaces, demonstrating scalability and robustness to increasing action-space complexity. We now explicitly highlight this point in the revised Introduction.

---

### Author Response · Authors · 2025-03-27
**Revised Manuscript**

We thank all reviewers for their thoughtful and constructive feedback. We have uploaded a revised version of the manuscript in response to your comments, with all additions highlighted for clarity. We have also addressed each reviewer's comments point by point in our individual responses. Please don’t hesitate to let us know if any aspect would benefit from further clarification or discussion.

---

> ### Comment · Action_Editor_rMwG · 2025-03-27
>
> Action editor, just checking in here.
>
> Authors: thanks for the responses and the revision.
>
> Reviewers: this is your opportunity to engage with each other and with the authors with the goal of gathering enough information to make an informed, confident decision. Please look over the authors' comments and the new manuscript and take the opportunity to follow up with any additional questions or unresolved concerns.

---

### Decision · Action_Editor_rMwG · 2025-04-29

**Recommendation:** Accept as is

**Comment:**

The reviewers agree that the revised manuscript is clear, sound, and interesting and does not require any revision to be published in TMLR.

**Audience:**

Yes, definitely. The paper offers improved understanding and practical guidance for RL in large/continuous action spaces. Currently policy-gradient (and related) approaches are dominant in such problems, but they come with their own tradeoffs and limitations. Progress toward making value-based methods an option in this type of problem is likely to be of interest to many in the RLC community.

**Claims And Evidence:**

Though there were initial questions about whether the core claims were well-defined (let alone well-supported), the reviewers agree that the revised manuscript has addressed those concerns. The consensus is that the paper now clearly states its claims and sufficiently supports them with empirical evidence.

---

> ### Author Response · Authors · 2025-06-03
> **Thank you!**
>
> We sincerely appreciate the thoughtful and high-quality reviews, which greatly improved the clarity and overall quality of our manuscript. We are grateful for the time and effort the reviewers and the action editor invested in evaluating our work.
>
> We are very pleased with the positive outcome and have now submitted the camera-ready version of our paper.
>
> Thank you again!